# On the flexibility of the cellular amination network in *E coli*

**Helena Schulz-Mirbach[1], Alexandra Müller[1], Tong Wu[1], Pascal Pfister[2], Selçuk Aslan[1], Lennart Schada von Borzyskowski[2,3], Tobias J Erb[2,4], Arren Bar-Even[1], Steffen N Lindner[1,5]***

[1]Max Planck Institute of Molecular Plant Physiology, Potsdam, Germany; [2]Max Planck Institute for Terrestrial Microbiology, Marburg, Germany; [3]Institute of Biology Leiden, Leiden University, Leiden, Netherlands; [4]Center for Synthetic Microbiology (SYNMIKRO), Marburg, Germany; [5]Department of Biochemistry, Charité – Universitätsmedizin Berlin, Berlin, Germany

**Abstract** Ammonium ($NH_4^+$) is essential to generate the nitrogenous building blocks of life. It gets assimilated via the canonical biosynthetic routes to glutamate and is further distributed throughout metabolism via a network of transaminases. To study the flexibility of this network, we constructed an *Escherichia coli* glutamate auxotrophic strain. This strain allowed us to systematically study which amino acids serve as amine sources. We found that several amino acids complemented the auxotrophy either by producing glutamate via transamination reactions or by their conversion to glutamate. In this network, we identified aspartate transaminase AspC as a major connector between many amino acids and glutamate. Additionally, we extended the transaminase network by the amino acids β-alanine, alanine, glycine, and serine as new amine sources and identified D-amino acid dehydrogenase (DadA) as an intracellular amino acid sink removing substrates from transaminase reactions. Finally, ammonium assimilation routes producing aspartate or leucine were introduced. Our study reveals the high flexibility of the cellular amination network, both in terms of transaminase promiscuity and adaptability to new connections and ammonium entry points.

**\*For correspondence:**
Lindner@mpimp-golm.mpg.de

**Competing interest:** The authors declare that no competing interests exist.

## Editor's evaluation

This article investigates the non-canonical ammonium metabolism in *E. coli*, exploring possible ways by which *E. coli* is able to assimilate nitrogen. The work reveals that some amination reactions can be reversed and redirected with strain engineering, thus introducing the novel concept of a cellular amination network (reversible nitrogen transfer). In addition to furthering a fundamental understanding of metabolism, this work provides a modular platform for the metabolic engineering of ammonium assimilation and dissimilation, which has potential for the industrial bioproduction of nitrogenous compounds.

## Introduction

Nitrogen is essential for all forms of life as it is part of 75% of the cell's building blocks (mainly in proteins and nucleic acids) (*Milo and Phillips, 2015*). The conversion of atmospheric dinitrogen ($N_2$) to ammonia ($NH_3$) by diazotrophic bacteria or industrially by the Haber–Bosch process is essential to make it available for assimilation by plants and other organisms to produce nitrogenous compounds.

While carbon fixation has evolved several times, resulting in versatile naturally occurring ways of carbon fixation (*Löwe and Kremling, 2021*), the introduction of ammonium ($NH_4^+$, protonated form of ammonia) into the building blocks of life is similar in all organisms and limited to the fixation

**eLife digest** Nitrogen is an essential part of many of the cell's building blocks, including amino acids and nucleotides, which form proteins and DNA respectively. Therefore, nitrogen has to be available to cells so that they can survive and grow. In nature, some microorganisms convert the gaseous form of nitrogen into ammonium, which then acts as the nitrogen source of most organisms, including bacteria, plants and animals. Once cells take up ammonium, it is 'fixed' by becoming part of an amino acid called glutamate, which has a so-called 'amine group' that contains a nitrogen. Glutamate then becomes the central source for passing these amines on to other molecules, distributing nitrogen throughout the cell.

This coupling between ammonium fixation and glutamate production evolved over millions of years and occurs in all organisms. However, the complete metabolic network that underlies the distribution of amines remains poorly understood despite decades of research. Furthermore, it is not clear whether ammonium can be fixed in a way that is independent of glutamate.

To answer these questions, Schulz-Mirbach et al. used genetic engineering to create a strain of the bacterium *E. coli* that was unable to make glutamate. These mutant cells could only grow in the presence of certain amino acids, which acted as alternative amine sources. Schulz-Mirbach et al. found that enzymes called transaminases, and one called AspC in particular, were required for the cells to be able to produce glutamate using the amine groups from other amino acids. Notably, Schulz-Mirbach et al. showed that AspC, which had previously been shown to use an amino acid called aspartate as a source of amine groups, is indispensable if the cell is to use the amine groups from other amino acids – including histidine, tyrosine, phenylalanine, tryptophan, methionine, isoleucine and leucine.

Schulz-Mirbach et al. also discovered that if they engineered the *E. coli* cells to produce transaminases from other species, the repertoire of molecules that the cells could use as the source of amines to generate glutamate increased. In a final set of experiments, Schulz-Mirbach et al. were able to engineer the cells to fix ammonium by producing aspartate and leucine, thus entirely bypassing the deleted routes of glutamate synthesis. These data suggest that fixing ammonium and distributing nitrogen in *E. coli* can be very flexible.

The results from these experiments may shed light on how cells adapt when there is not a lot of ammonium available. Moreover, this study could advance efforts at metabolic engineering, for example, to create molecules through new pathways or to boost the production of amino acids needed for industrial purposes.

of ammonium at the node between 2-ketoglutarate, glutamate, and glutamine (*Figure 1*). Here, three cooperating canonical enzymes assimilate ammonium in two distinct ways. In the first reaction, glutamate is the direct product of glutamate dehydrogenase (*gdhA*, GDH). In the second pathway, the combined activity of glutamine synthetase (*glnA*, GS) and glutamate synthase (glutamine 2-ketoglutarate aminotransferase, *gltBD*, GOGAT) fixes another ammonium and converts glutamate into glutamine, which then donates one amine to 2-ketoglutarate to form two glutamate molecules (*Helling, 1994*; *Kumar and Shimizu, 2010*; *Figure 1*). To make all essential amino acids and other aminated compounds, glutamate and glutamine then donate their amines to specific keto acids or other amino acid precursors in reversible transferase reactions. As ammonium enters metabolism solely via these two routes, all cellular nitrogen is provided by either glutamate (75%) or glutamine (25%) (*Yang et al., 2018*).

Besides the glutamate biosynthesis node, alternative entry points for ammonium theoretically exist, for example, alanine dehydrogenase or aspartate ammonia lyase, but these are not relevant for ammonium assimilation (*Kim and Hollocher, 1982*). Evolution has developed a system for ammonium assimilation, which is controlled by its intracellular availability. The ATP investment driving GS activity makes amination reactions favorable even at low ammonium concentrations. Additionally, most of the GS orthologs have evolved kinetic parameters optimized for lower ammonium concentrations with an apparent $K_M$ of 0.1 mM for ammonium (*Reitzer, 2014*). At high ammonium concentrations, the NADPH-dependent (and thus energetically cheaper) GDH allows more efficient ammonium assimilation than GS (*Reitzer, 2014*). This metabolic switch might explain the prominence of glutamate-based ammonium assimilation as opposed to other ammonia entry points in nature. To generate the

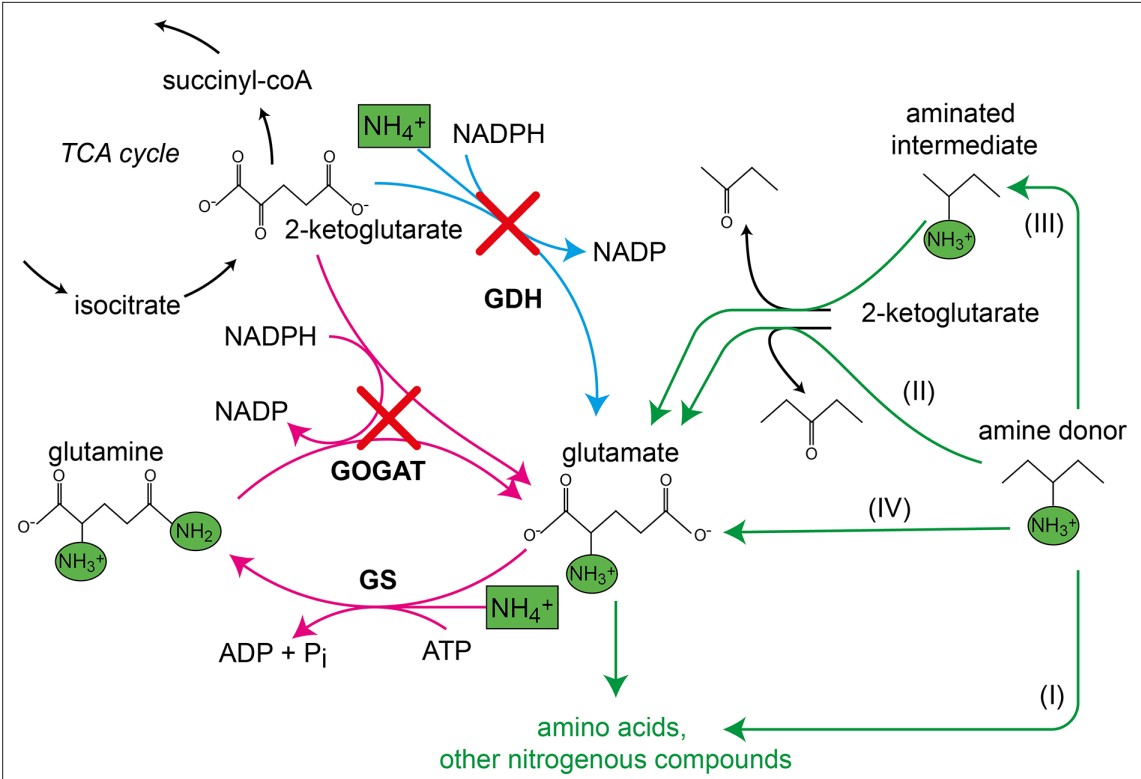

**Figure 1.** Canonical ammonium assimilation via glutamate dehydrogenase (GDH) (blue arrow) or glutamine synthetase (GS) and glutamine 2-ketoglutarate aminotransferase (GOGAT) (pink arrows). One pathway for ammonium (green boxes) assimilation is the amination of the tricarboxylic acid (TCA) cycle intermediate 2-ketoglutarate by GDH to form glutamate (blue arrow). A second pathway requires joint action of GS and GOGAT, which first aminate glutamate to form glutamine (GS), which donates one amine to 2-ketoglutarate (GOGAT) to form two glutamate molecules, which further provide amines (green circles) for biosynthesis of amino acids and other nitrogenous compounds (pink arrows). Growth of the glutamate auxotrophic (glut-aux) strain deleted in GDH and GOGAT (red crosses) by a supplemented amine source is possible via the following mechanisms. The amine donor either (I) replaces glutamate as an amine source for the production of amino acids and nitrogenous compounds, (II) donates an amine to 2-ketoglutarate to form glutamate, (III) is converted to an intermediate donating an amine to 2-ketoglutarate, or (IV) is metabolically converted into glutamate. Green arrows indicate these cases.

NADPH required for GDH, *Escherichia coli* mostly uses the membrane-bound proton-translocating transhydrogenase (PntAB) (*Sauer et al., 2004*). This enzyme exploits the proton motif force to drive proton translocation from NADH to NADP+ (*Spaans et al., 2015*), and thereby indirectly competes with ATP synthesis. Therefore, when growing under high ammonium concentrations, growth of this microorganism might benefit from ammonium assimilation via NADH-dependent dehydrogenases.

Following these thoughts, we tried to assess whether alternative routes for ammonium assimilation can arise from the metabolic network of *E. coli*. For this purpose, we systematically investigated the flexibility of the amination network in a glutamate auxotrophic – and hence ammonium assimilation deficient – *E. coli* strain. This study provides fundamental knowledge on the plasticity of ammonium metabolism in *E. coli* and moreover addresses industrial interests by providing a versatile bacterial *chassis* for screening and optimization of ammonium assimilation and transamination reactions.

## Results

### Only some amino acids serve as an amine source

To study the flexibility of *E. coli*'s cellular amination network, we first generated a strain in which both canonical ammonia assimilation routes were disrupted. Accordingly, we deleted the genes encoding GDH (*gdhA*) and GOGAT (*gltBD*), which are responsible for 2-ketoglutarate amination under high and low ammonia concentrations, respectively (*Helling, 1994*; *Kumar and Shimizu, 2010*; *Figure 1*). The resulting glutamate auxotrophic strain (glut-aux, Δ*gdhA* Δ*gltBD*) was not able to grow in minimal

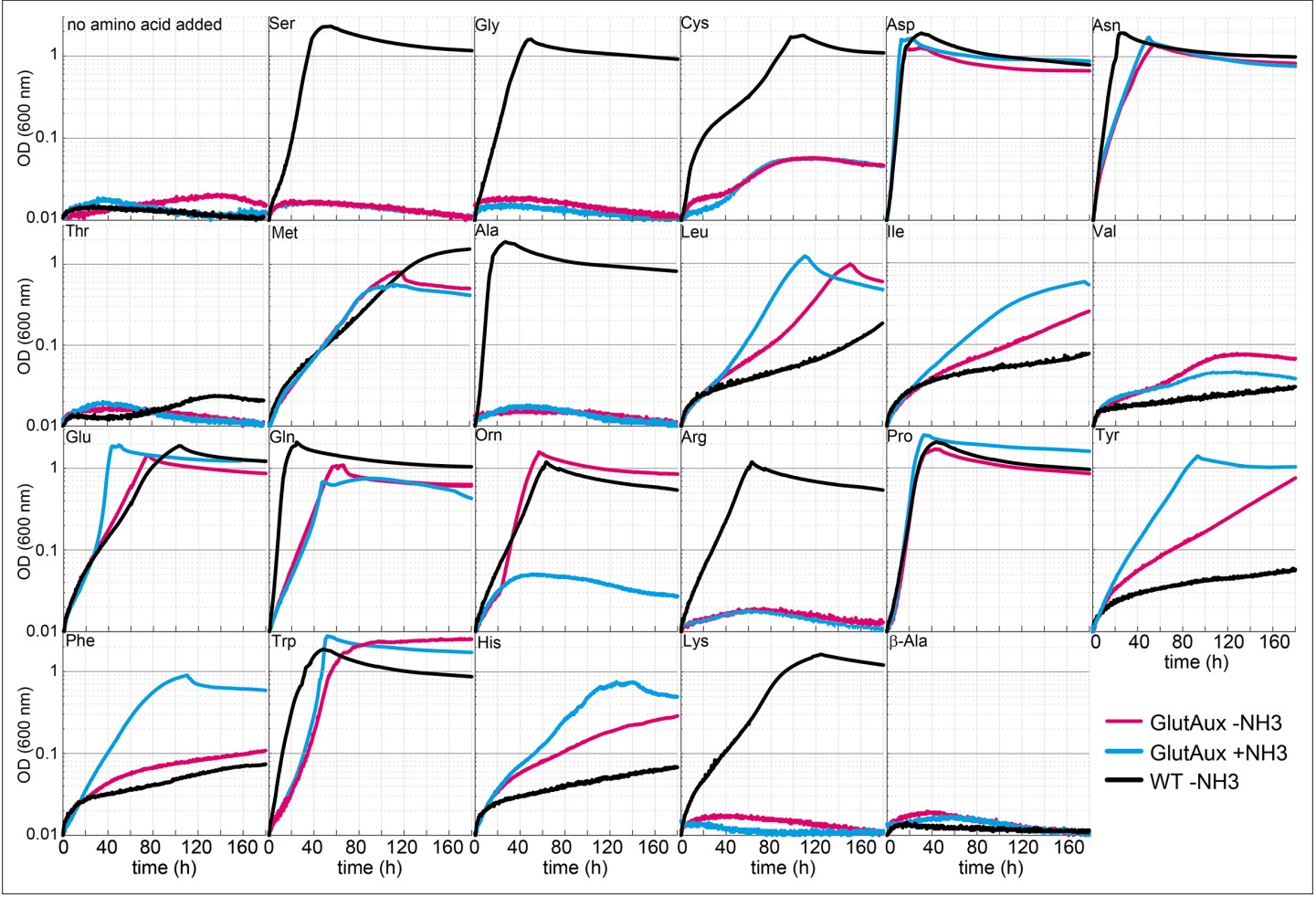

**Figure 2.** Identification of amino acids that rescue growth of the glutamate auxotrophic (glut-aux) strain. The glut-aux strain was grown in M9 medium with (blue line) or without ammonium (magenta line) and 20 mM glycerol as carbon source. *E. coli* wildtype (WT) was grown in M9 medium without ammonium (black line) and 20 mM glycerol as carbon source. 5 mM of the indicated amino acids or no amino acid as negative control were supplemented to test whether they can serve as an amine source (glut-aux strain) or an ammonium source (WT). Data shows representative growth as observed from triplicate repeats with errors <5%.

The online version of this article includes the following figure supplement(s) for figure 2:

**Figure supplement 1.** Summary of growth rescue experiment (*Figure 2*) with wildtype (WT) and glutamate auxotrophic (glut-aux) strain.

medium with ammonium as the sole nitrogen source unless an amine group donor, such as glutamate, was provided in the medium (*Figure 2*). While D-glutamate-requiring *E. coli* mutants had been previously characterized (*Dougherty et al., 1993*), our strain lacking glutamate biosynthetic pathways represents the first strain for a systematic investigation of the metabolic network for amine distribution. To this end, we first tested whether other amino acids can replace glutamate as an amine source allowing growth of the glut-aux. We therefore characterized growth of the glut-aux strain when supplemented with one of the proteinogenic amino acids. We note that this growth experiment is different from the experiments commonly described in the literature, where amino acids were added to the medium without ammonium to serve as the sole nitrogen source (*Neidhardt et al., 1996*). In these experiments, the metabolic degradation of the amino acids to release ammonia suffices to enable growth. Conversely, for the glut-aux strain, growth complementation through the supplemented amino acids as an amine source must follow one of these options: (i) substitute glutamate as an amine source for the production of other nitrogenous compounds; (ii) serve as an amine group donor for 2-ketoglutarate to generate glutamate as an amine source; (iii) be further converted to compounds to serve as an amine group donor for 2-ketoglutarate amination; or (iv) be directly converted to glutamate (*Figure 1*). In the first three cases, the amination network of the cell needs

to be flexible enough to adapt to different directionalities of at least some of the transamination reactions. As utilization of amino acids as amine source in the glut-aux strain might be dependent on the nitrogen-regulated (Ntr) response (*Reitzer, 2003*), we performed growth experiments with and without ammonia in the medium (light blue and magenta lines in *Figure 2*). As a control, we repeated the classical experiments of testing each amino acid as the sole nitrogen source with a wildtype strain. Here, amino acids are not required to directly donate their amine group but can rather support growth by releasing ammonia through their degradation (black lines in *Figure 2*). Besides the proteinogenic amino acids, we also sought to investigate whether the naturally occurring, non-proteinogenic amino acids β-alanine and ornithine could rescue the growth of the glut-aux strain. These amino acids are derived from aspartate or glutamate, respectively, which, due to the presence of respective transaminases, were amine sources for both glut-aux and wildtype. Thus, β-alanine and ornithine are directly connected to the amination network. In our experiments, we chose to use glycerol rather than glucose as a carbon source to avoid interference of glucose-dependent inducer exclusion and catabolite repression with amino acid utilization. Moreover, previous studies showed that *E. coli* acquires higher growth rates with glycerol as carbon and energy source when amino acids are provided as amine sources (*Bren et al., 2016*). To minimize selective pressure in the preculture and avoid accumulation of mutations prior to the experiment, all experiments were inoculated from cultures grown in glycerol minimal medium supplemented with 5 mM aspartate, the amino acid immediately supporting fastest growth of the glut-aux strain (*Figure 2*, *Supplementary file 1*).

We found that only some amino acids rescued growth of the glut-aux strain and observed a high variety in growth rate upon feeding of different amino acids (*Figure 2*, *Supplementary file 1*). This generally correlated with the existence of known transaminase enzymes that enable glutamate production from the respective amino acids in *E. coli* (rescue mechanisms II and III, *Figure 1*). For example, aspartate, leucine, and tyrosine serving as cellular amine donor for glutamate generation from 2-ketoglutarate (rescue mechanism II, *Figure 1*) could be attributed to the activity of aspartate transaminase (AspC), tyrosine transaminase (TryB), and branched-chain amino acid transaminase (IlvE) (*Gelfand and Steinberg, 1977*). As these transaminases display considerable cross-reactivity (*Gelfand and Steinberg, 1977*; *Inoue et al., 1988*), each of these three amino acids might support the production of the others directly, without the need for glutamate as an amine donor. However, glutamate here still likely serves as the primary amine donor for most cellular nitrogenous compounds (*Yang et al., 2018*). Hence, AspC, TryB, and IlvE must also be fully reversible in the glut-aux strain to aminate 2-ketoglutarate to glutamate. While transaminases are generally reversible enzymes, their ability to effectively operate reversibly in vivo is not trivial as the [glutamate]/[2-ketoglutarate] ratio is very high (above 100) under physiological conditions (*Bennett et al., 2009*), making the reverse amine transfer onto 2-ketoglutarate to form glutamate thermodynamically challenging. Since the glut-aux strain grew with several amino acids as an amine source, we conclude that the cellular amination network must be sufficiently flexible to accept amine sources other than glutamate despite the potential thermodynamic barriers. However, since metabolite concentrations and especially the [glutamate]/[2-ketoglutarate] ratio in the glut-aux strain likely differ from that in a wildtype strain, the reverse transaminase activities might be favored in this synthetic strain. Of the supplied non-proteinogenic amino acids, ornithine rescued growth of the glut-aux strain in the absence of NH₄Cl (and was used as a nitrogen source by the WT). In *E. coli*, ornithine is degraded to putrescine, which is used by the transaminase PatA to aminate 2-ketoglutarate to glutamate (*Schneider et al., 2013*; *Schneider and Reitzer, 2012*). Expression of this degradation pathway is induced by nitrogen limitation (*Zimmer et al., 2000*) and allows growth with ornithine (rescue mechanism III, *Figure 1*). In contrast, β-alanine was neither an amine source of the glut-aux strain nor served as a nitrogen source for the WT (*Figure 2*).

To prove amine transfer from the provided amino acids, we cultivated the glut-aux strain with 5 mM of one of five (unlabeled) representative amino acids that can serve as an amine donor – glutamate, proline, aspartate, tryptophan, and leucine – in medium containing 20 mM $^{15}$N-ammonium. We subsequently measured the $^{15}$N labeling in some proteinogenic amino acids ('Materials and methods') that represent a broad range of amino acids. We chose alanine (A), the direct product of pyruvate amination, phenylalanine (F) as an example of an aromatic amino acid, leucine (L), representing branched-chain amino acids, methionine (M) and threonine (T), representing aspartate-family amino acids, proline (P) as part of the glutamate-family amino acids, serine (S), representing serine-family amino

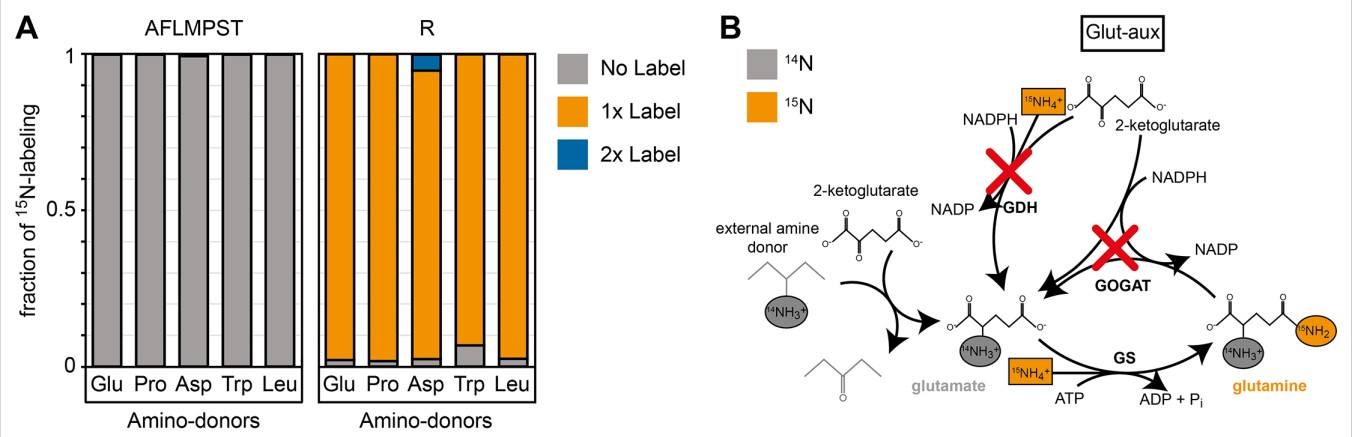

**Figure 3.** $^{15}$N labeling confirms amine assimilation from the supplied amino acid in the glutamate auxotrophic (glut-aux) strain. (**A**) The glut-aux strain was incubated in $^{15}$N-NH$_4$Cl M9 medium with 20 mM glycerol as carbon source. $^{15}$N labeling pattern in analyzed proteinogenic amino acids (single letter code) upon feeding with 5 mM of unlabeled amino acids glutamate, proline, aspartate, tryptophan, or leucine as amine donors (three-letter code). The labeling patterns of the amino acids A, F, L, M, P, S, and T were identical to the supplied amino acids and hence only a representative dataset is shown. Data represents means of triplicate measurements with errors <5%. (**B**) Schematic presentation of expected ammonium incorporation in glut-aux grown on M9 with $^{15}$N-NH$_4$Cl with 20 mM glycerol as carbon source and an unlabeled amino acid as ammonium source. Since the genes encoding glutamate dehydrogenase (GDH) and glutamine 2-ketoglutarate aminotransferase (GOGAT) are deleted in the glut-aux (red crosses), the glut-aux thus relies on the provided amino acid for biosynthesis of unlabeled (gray) glutamate. During glutamine biosynthesis, the glut-aux assimilates free ammonium (boxes) supplied as $^{115}$N-NH$_4$Cl to form once-labeled glutamine that is used for carbamoyl-phosphate biosynthesis and thus results in once-labeled arginine.

acids, and arginine (R), which, additionally to glutamate-derived amines, uniquely carries an amine from carbamoyl phosphate that originated from the δ-amino group of glutamine that was initially formed from free ammonium by glutamine synthetase. In these experiments, we found that most of the amino acids were completely unlabeled (*Figure 3A*), confirming that their amine group was transferred from the respective amine-donating amino acid rather than from free ($^{15}$N labeled) ammonia in the medium (*Figure 3B*). In these experiments, arginine appeared single-labeled (R, *Figure 3*) as one of its nitrogen atoms originates from the amide nitrogen of glutamine that is derived from ammonia fixed by GS activity (*Figure 3B*). Overall, these results confirm that the amino acids added to the medium were the only amine sources allowing growth of the glut-aux strain, rather than allowing amination of amino acid-derived backbones with free ammonium in the medium.

To further validate that growth of the glut-aux strain is limited by the supply of amine groups from the amino acids provided in the medium, we cultivated it using different concentrations of amino acids (*Figure 4*). As expected, we found that biomass yield, as indicated by the maximal OD$_{600}$, directly correlated with the concentration of the supplemented amino acid. All amino acids showed the same dependency of biomass yield on concentration, with the exception of ornithine, which supported roughly double the yield for each concentration. This is in line with the fact that transaminases can transfer both amine groups from the ornithine degradation intermediate putrescine to 2-ketoglutarate to form glutamate (*Prieto-Santos et al., 1986*) (rescue mechanism III, *Figure 1*). As this ornithine degradation pathway is induced by nitrogen starvation (*Schneider et al., 2013*; *Schneider and Reitzer, 2012*), growth of the glut-aux strain with ornithine as amine group donor was observed only when ammonia was omitted in the medium. The correlation of maximal OD$_{600}$ to amino acid concentration confirms that amine supply from the amino acid limits biomass yield in the glut-aux strain in the same manner for all tested amino acids. To our surprise, glutamate was not the amine donor supporting fastest growth of the glut-aux strain. Even proline, which, in order to donate its amine group, needs to be converted to glutamate (rescue mechanism IV, *Figure 1*), supported faster growth (*Figure 4*). This, together with the fact that the growth rate increased proportionally with the glutamate concentration, indicates that glutamate uptake is limiting growth of the glut-aux strain.

## The cellular amination network is highly promiscuous

To investigate the contribution of different enzymes to the use of amino acids as an amine source, we decided to analyze the effects of several gene deletions. We specifically deleted genes encoding

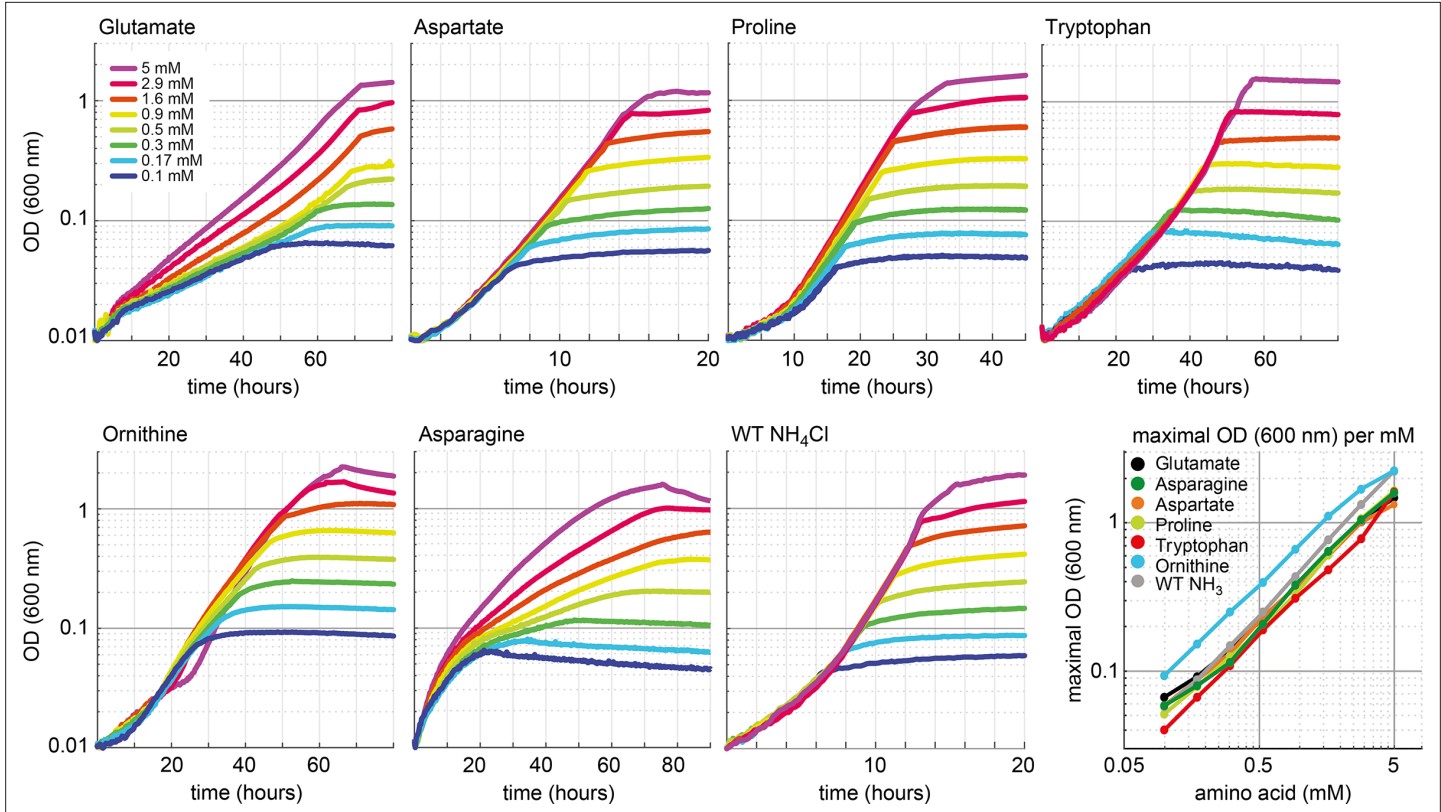

**Figure 4.** Growth dependency of the glutamate auxotrophic (glut-aux) strain on amino acid concentration. Cells were grown in ammonium-free M9 medium with 20 mM glycerol and the indicated concentrations of the amino acids glutamate, aspartate, proline, tryptophan, ornithine, and asparagine. As a comparison, wildtype (WT) was grown in ammonium-free M9 medium with 20 mM glycerol but with NH₄Cl concentrations similar to the amino acid concentrations used. Data shows representative growth as observed from triplicate repeats with errors <5%.

proteins essential for the certain glutamate-forming degradation pathways based on available documentation in the database Ecocyc (*Keseler et al., 2021*). First, we explored the use of proline as an amine source (*Figure 2*). We did not expect this amino acid to donate its amine group directly, but rather to be metabolized by PutA, encoding for a bifunctional flavoenzyme with proline dehydrogenase and 1-pyrroline-5-carboxylate dehydrogenase activities (*Moxley et al., 2014*), to glutamate (*Frank and Ranhand, 1964*), which would serve as an amine source (rescue mechanism IV, *Figure 1*).

Indeed, upon deletion of *putA*, growth of the WT and glut-aux strain on proline as the sole ammonium source was abolished (*Figure 5A*). This confirmed that proline could only support growth via its native degradation pathway and was unable to serve as an amine source via a different pathway. We then focused on amino acids that are expected to directly donate their amine groups. For example, methionine, which could serve as an amine source, was previously found to be the main substrate for only a single transaminase: YbdL (*Dolzan et al., 2004*). However, deletion of *ybdL* in the Δ*gdhA* Δ*gltBD* strain did not affect growth with methionine as an amine donor (*Figure 5—figure supplement 1*), suggesting that the physiological contribution of this transaminase to the use of methionine as an amine source is negligible. YbdL was also shown to be the only transaminase able to efficiently accept histidine and phenylalanine (*Dolzan et al., 2004*; *Inoue et al., 1988*). Yet, the glut-aux Δ*ybdL* strain did not show any growth retardation when using histidine or phenylalanine as an amine donor (*Figure 5—figure supplement 1*). Indeed, other transaminases are known to accept methionine and histidine, albeit at a substantially lower affinity and rate than their primary substrate, for example, aspartate transaminase AspC, tyrosine transaminase TyrB, and branched-chain amino acid transaminase IlvE (*Inoue et al., 1988*; *Mavrides, 1987*; *Powell and Morrison, 1978*). It therefore seems that the promiscuity of such transaminases enables effective use of methionine as an amine donor even in the absence of the enzyme preferring it as substrate. While previous studies already reported high enzymatic redundancy as a characteristic feature of transaminases (*Lal et al., 2014*), we aimed to

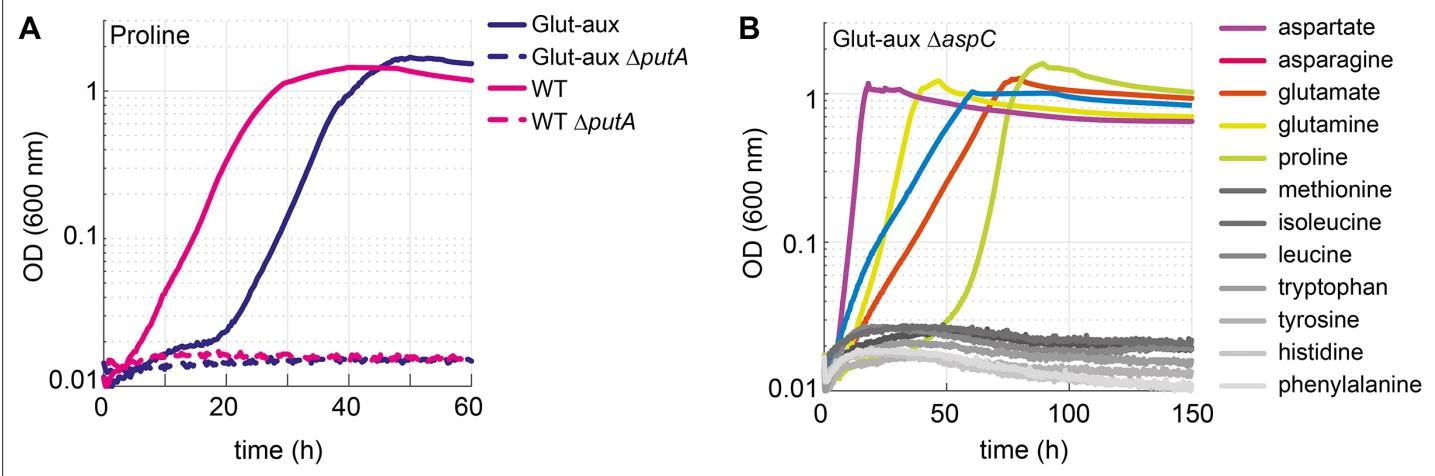

**Figure 5.** Proline is converted to glutamate to serve as an amine source. Aspartate transaminase (AspC) is responsible for the utilization of several amino acids. (**A**) Deletion of *putA* abolished growth with proline as an amine source in the glutamate auxotrophic (glut-aux) strain and as a nitrogen source in the wildtype (WT). (**B**) Deletion of *aspC* eliminates growth of the glut-aux with methionine, leucine, isoleucine, histidine, tyrosine, tryptophan, and phenylalanine as amine donors (gray lines). Experiments were carried out in M9 with ammonium containing 20 mM glycerol and 5 mM of the indicated amine sources. Data represents means of triplicates with <5% variation.

The online version of this article includes the following figure supplement(s) for figure 5:

**Figure supplement 1.** Deletion of *ybdL* does not alter growth of the glutamate auxotrophic (glut-aux) strain with phenylalanine, methionine, and histidine as amine sources.

further investigate the relevance of promiscuous transaminase activities for the utilization of external amine donors. Here, we focused on AspC, which is known to accept a range of amino acids (*Lal et al., 2014*). After constructing the glut-aux Δ*aspC* strain, we analyzed its growth on all previously tested 22 amino acids. We found that the glut-aux Δ*aspC* strain was unable to grow with histidine, tyrosine, phenylalanine, tryptophan, methionine, isoleucine, and leucine as amine donors, which all allowed growth of the glut-aux strain (gray lines in *Figure 2*, *Figure 5B*). This demonstrates promiscuity of AspC and reveals that, in its physiological context, this transaminase is even more versatile than previously reported. Although AspC had shown low specific transaminase activity with histidine, methionine, isoleucine, and leucine in vitro (*Hayashi et al., 1993*), our experiments suggest, however, that the enzyme is essential for the utilization of these amino acids as amine donors in vivo (rescue mechanisms II or III, *Figure 1*). Furthermore, TyrB and IlvE, which had shown higher specific in vitro activities with tyrosine, phenylalanine, and tryptophan compared to AspC (*Hayashi et al., 1993*), apparently did not complement the *aspC* deletion.

## Engineering utilization of alanine as an amine donor

Despite the presence of two glutamate-producing alanine transaminases (AlaA, AlaC) as well as multiple transaminases that can promiscuously accept alanine (*Kim et al., 2010*; *Peña-Soler et al., 2014*), alanine did not serve as an amine source for the glut-aux strain, regardless of the presence or absence of ammonia in the medium (*Figure 2*). Even upon overexpression of the alanine transaminase isozymes AlaA and AlaC, no growth with alanine as an amine donor was observed (*Figure 6A*). However, we tested whether prolonged incubation of the glut-aux strain with and without overexpression of either AlaA or AlaC on 20 mM glycerol +5 mM alanine might trigger the emergence of spontaneous mutants. Indeed, in this experiment two cultures of the glut-aux strain overexpressing AlaA could use alanine as an amine donor. These spontaneous mutants arose after a cultivation time of 110 and 133 hr, respectively. Glut-aux strains overexpressing AlaC or not overexpressing any alanine transaminases did not grow within the duration of the experiment (*Figure 6—figure supplement 1A*).

In the next step, we isolated single colonies on LB plates from the two AlaA overexpressing cultures. These colonies, upon transfer to M9 medium, grew immediately with 20 mM glycerol and 5 mM alanine, indicating that mutations had allowed their growth (*Figure 6A*). We sent two of the independently obtained isolates and the parent strain for genome sequencing. Analysis of the genome

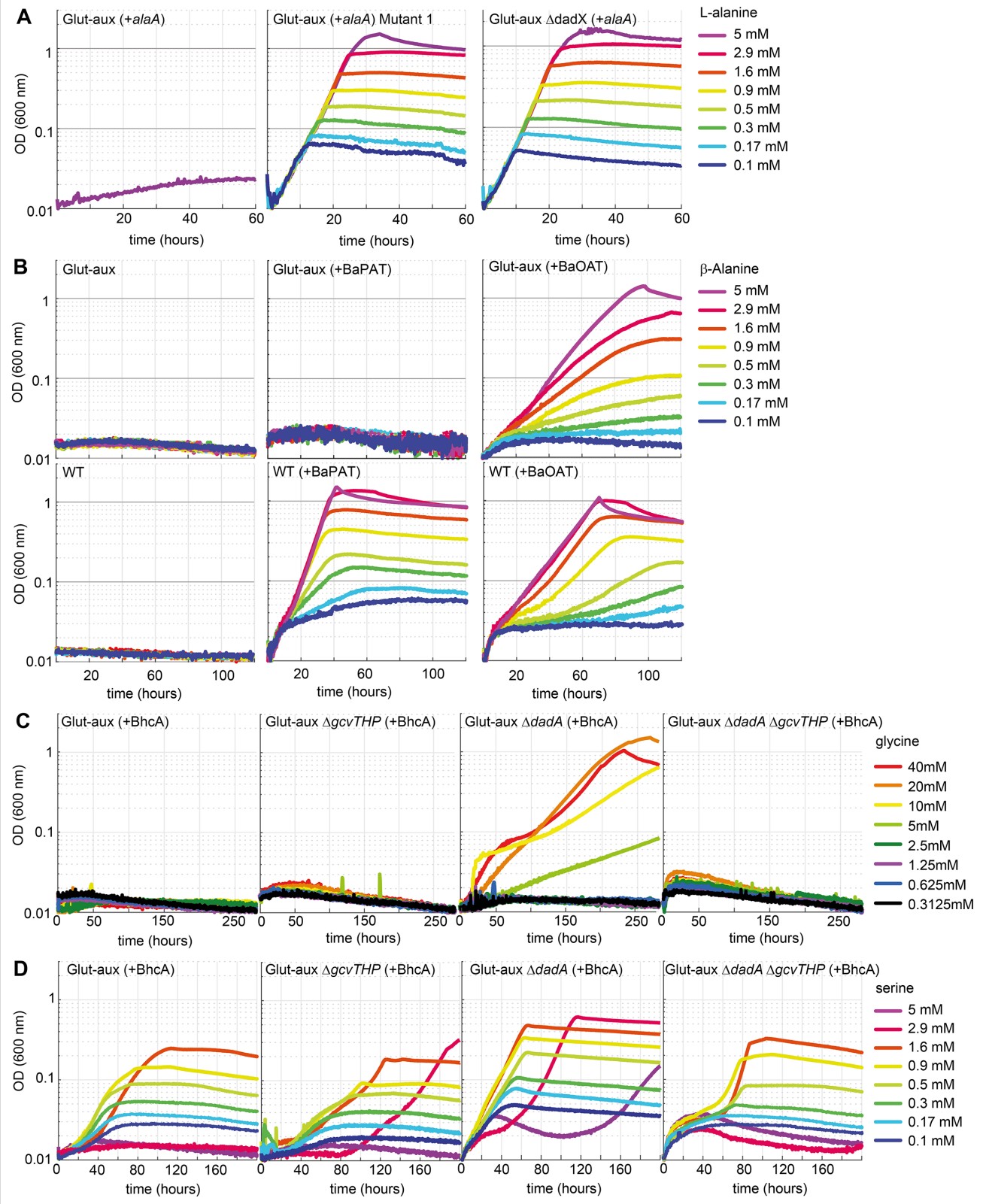

**Figure 6.** Glutamate auxotrophic (glut-aux) strains engineered to use alanine, β-alanine, glycine, and serine as an amine source. (**A**) DadX mutation or deletion and alanine transaminase overexpression allows alanine utilization as an amine source. Growth of the glut-aux strains on M9 medium with ammonium and 20 mM glycerol with 5 mM alanine as an amine donor. (**B**) Overexpression of β-alanine-2-ketoglutarate aminotransferase (BaOAT) enables amine donation from β-alanine in the glut-aux strain. Glut-aux strain and wildtype (WT) overexpressing β-alanine-pyruvate aminotransferase

*Figure 6 continued on next page*

*Figure 6 continued*

(BaPAT) or β-alanine-2-ketoglutarate aminotransferase (BaOAT) where grown in media containing no N-source (WT) or ammonium chloride (glut-aux), 20 mM glycerol, and the indicated β-alanine concentrations. (**C**) DadA deletion and BhcA overexpression allow glycine utilization as an amine donor. Growth of the glut-aux strains + BhcA on M9 with ammonium and 20 mM glycerol with indicated glycine concentrations. Data represents triplicate measurements with <5% variation. (**D**) BhcA overexpression allows use of serine as an amine source, which is improved by additional deletion of *dadA*. Growth of the glut-aux strains + BhcA on M9 with ammonium and 20 mM glycerol with indicated serine concentrations. All data represents triplicate measurements with <5% variation.

The online version of this article includes the following figure supplement(s) for figure 6:

**Figure supplement 1.** Mutations in the *dadX* locus allow transaminase-based use of alanine as an amine source.

**Figure supplement 2.** DadX mutation or deletion and alanine transaminase overexpression allows alanine utilization as an amine source.

**Figure supplement 3.** $^{15}$N labeling experiments confirm incorporation of the amino group of β-alanine into proteinogenic amino acids (single-letter code).

**Figure supplement 4.** Transaminase overexpression is essential for the use of serine or glycine as an amine donor in the glutamate auxotrophic (glut-aux) strain.

**Figure supplement 5.** Michaelis–Menten kinetics of DadA and BhcA for the selected substrates.

**Figure supplement 6.** Schematic presentation of glycine and serine utilization by the glutamate auxotrophic (glut-aux) strain Δ*dadA* expressing BhcA.

**Figure supplement 7.** $^{15}$N labeling confirms amine donation from serine or glycine in glutamate auxotrophic (glut-aux) ΔdadA + BhcA.

sequencing results revealed that both isolates had mutations in the alanine racemase gene *dadX* (***Supplementary file 2***, ***Figure 6—figure supplement 1B***). The mutations were either a duplication of a 27 bp region from nucleotide 151 to nucleotide 177 (mutant 1) or a 1 bp deletion at nucleotide position 821 (of the 1071 nucleotide gene) (mutant 2). In both cases, the mutations resulted in a frameshift, indicating a loss of function of DadX. Alanine racemase diverts flux from L-alanine to D-alanine, which is either used for cell wall biosynthesis (***Walsh, 1989***) or degraded to pyruvate via D-alanine:quinone oxidoreductase (DadA) (***Franklin and Venables, 1976***). The absence of DadX activity likely reduces flux into this alanine sink, resulting in higher alanine availability, which might be responsible for growth rescue in the mutants. To verify this hypothesis, we deleted *dadX* in the evolved strain, as well as in the parental glut-aux strain. Both strains (with overexpression of AlaA) immediately grew with alanine as an amine donor (***Figure 6A***, ***Figure 6—figure supplement 2***), verifying that reducing an important intracellular alanine sink allowed the overexpressed transaminase to use alanine as an amine donor to effectively sustain growth. Noteworthy, a deletion of DadA in the glut-aux strain overexpressing AlaA did not restore growth with alanine as an amine donor, indicating that the reversible conversion of L-alanine to D-alanine is already sufficient to reduce alanine availability below a critical level, where it cannot support enough flux via AlaA for cellular growth. Note that *dadX* is not essential as *E. coli* possesses additional genes that are able to provide sufficient D-alanine for cell wall biosynthesis (***Kang et al., 2011***; ***Wild et al., 1985***).

## Engineering the utilization of β-alanine as an amine donor

The non-proteinogenic amino acid β-alanine is neither a suitable amine donor for the glut-aux strain nor an N-source for the wildtype (***Figure 2***). However, the transamination product of β-alanine, malonate semialdehyde, plays a pivotal role as an intermediate for the biosynthesis of some biotechnologically important products, for example, 3-hydroxypropionate (***Vidra and Németh, 2018***). Hence, engineering the use of β-alanine as an amine donor for the glut-aux strain could potentially open new routes for selective 3-hydroxypropionate production. In order to test whether β-alanine usage can be engineered, two different β-alanine transaminases were overexpressed in the glut-aux and WT strains. These were the β-alanine-2-ketoglutarate aminotransferase from *Saccharomyces kluyveri* (BaOAT, UniProt ID A5H0J5) and the β-alanine-pyruvate aminotransferase from *Pseudomonas aeruginosa* (BaPAT, UniProt ID Q9I700). Overexpression of either BaPAT or BaOAT allowed the wildtype strain to grow using β-alanine as the sole source of ammonia (***Figure 6B***). The enzymes transfer the amine group from β-alanine onto the ketoacids pyruvate or 2-ketoglutarate to generate alanine and glutamate, respectively. Both alanine and glutamate serve as N-sources for the wildtype (***Figure 2***). The glut-aux strain however was only able to grow with β-alanine as an amine donor when BaOAT, but not BaPAT, was overexpressed (***Figure 6B***), which is in good agreement with our previous results that glutamate, but not alanine, can immediately serve as an amine donor in the glut-aux strain (***Figure 2***).

To further confirm that the growth of the glut-aux + BaOAT strain was rescued by directly using β-alanine via the proposed transamination reaction catalyzed by BaOAT, we conducted a nitrogen-tracing experiment. The medium contained $^{15}$N-labeled ammonium together with 20 mM glycerol and 5 mM β-alanine. As expected, the majority of amino acids did not show any $^{15}$N labeling (*Figure 6—figure supplement 3*), confirming that their amino group originated from β-alanine, rather than from ammonium. The only exception were arginine and glutamine that were single-labeled, with the $^{15}$N label originating from glutamine synthetase activity (see above).

## Overexpression of a glycine-oxaloacetate transaminase allows growth of the glut-aux strain with glycine after deletion of a glycine sink

Like alanine, glycine did not serve as an amine source in the glut-aux strain (*Figure 2*). However, glycine and serine are central metabolites of the reductive glycine pathway, which was previously shown to enable formatotrophic growth of *E. coli*. The key reaction of the pathway is the formation of glycine from free ammonium, $CO_2$, and 5,10-methylene-THF via the reductive activity of the glycine cleavage system (*Claassens et al., 2022*; *Kim et al., 2020*). Transamination of glycine or serine instead of deamination can improve efficiency of the formate assimilation pathway (*Claassens et al., 2022*). As *E. coli* lacks a glycine transaminase, this amino acid cannot be directly generated by transamination of the respective ketoacid (glyoxylate). Instead, it is obtained from serine via the serine hydroxymethyl transferase reactions. To engineer the usage of glycine as an amine donor, we overexpressed the glycine-oxaloacetate transaminase from *Paracoccus denitrificans* (BhcA, UniProt ID A1B8Z3) in the glut-aux strain to form aspartate, which we have shown to support fast growth of the glut-aux strain (*Figure 2*).

However, growth of the glut-aux strain overexpressing BhcA was not restored upon supplying the medium with glycine (*Figure 6C*, *Figure 6—figure supplement 4*). We speculated that, similar to our results with alanine as an amine donor described above, the provided glycine is further diverted into a sink. Hence, we directed our efforts towards the potential promiscuous activity of the D-alanine:quinone oxidoreductase DadA with glycine and the glycine cleavage system. DadA has been reported to be active with a variety of D-amino acids (*Wild and Klopotowski, 1981*) and might potentially use glycine as a substrate as well. Indeed, the deletion of *dadA* allowed the strain to grow slowly with glycine as an amine donor (*Figure 6C*), at first indicating glycine removal via promiscuous activity of DadA. However, in vitro, DadA showed only a very low specific activity with glycine (*Figure 6—figure supplement 5A*) and an apparent $K_M > 500$ mM for this amino acid (*Supplementary file 3*), which strongly suggested that the catalytic efficiency ($k_{cat}/K_M$ of 0.275 M$^{-1}$ s$^{-1}$) of the enzyme (*Figure 6—figure supplement 5A*, *Supplementary file 3*) was insufficient to provide a strong glycine sink in vivo. Notably, DadA was previously reported to efficiently convert D-serine (*Wild and Klopotowski, 1981*), the product of alanine racemase Alr, which also acts on L-serine besides L-alanine (*Ju et al., 2005*). In vitro measurements of DadA confirmed a higher catalytic efficiency ($k_{cat}/K_M$ of 5.71 M$^{-1}$ s$^{-1}$) of DadA with D-serine (*Figure 6—figure supplement 5A*) than previously measured with glycine. BhcA exhibits much better kinetic properties with serine (tested with glyoxylate as the acceptor [$K_M$ 2.1 mM; $k_{cat}$ 8.8 s$^{-1}$]) than with glycine (*Schada von Borzyskowski et al., 2019*). We thus concluded that glycine needed to be converted to serine via combined activities of the glycine cleavage system (GCV), cleaving one glycine molecule to ammonium, $CO_2$, and 5,10-methylene-tetrahydrofolate, and serine hydroxymethyl transferase subsequently condensing the latter with a second glycine molecule to form serine. This pathway is used by *E. coli* when utilizing glycine as the sole source of nitrogen (*Newman et al., 1976*). Supporting this hypothesis, the deletion of the GCV subunits *gcvTHP* in the glut-aux strain overexpressing BhcA did not allow growth with glycine. Moreover, deletion of *gcvTHP* in the glut-aux Δ*dadA* strain overexpressing BhcA abolished growth with glycine (*Figure 6C*), providing strong evidence that glycine is converted to serine to utilize it as an amine donor (*Figure 6—figure supplement 6*). Moreover, compared to other amine donors tested the growth supported by glycine was very slow (stationary phase reached after 200 hr).

Next, we tested whether serine (not an amine source for the glut-aux strain; *Figure 2*) served as an amine donor for the glut-aux strain overexpressing BhcA. In good agreement with the conclusion of our glycine trials, we found that the growth of the glut-aux, glut-aux Δ*dadA*, glut-aux Δ*gcvTHP*, and glut-aux Δ*dadA* Δ*gcvTHP* strains (all overexpressing BhcA) was restored upon addition of low serine concentrations (*Figure 6D*). In our growth experiments, serine concentrations above 1.7 mM seemed

to be toxic to the strain, which can be explained by the inhibition of isoleucine and aromatic amino acid biosynthesis by serine-derived hydroxypyruvate (*Hama et al., 1990*). The glut-aux Δ*dadA* reached the highest $OD_{600}$ on all tested serine concentrations, indicating that also here deletion of DadA is beneficial as it removes a sink for either glycine or serine directly (*Figure 6D*). To further demonstrate serine transamination catalyzed by BhcA with oxaloacetate as amine acceptor, we conducted in vitro measurements with purified BhcA. The enzymatic coupling assay indeed confirmed amine transfer from serine to oxaloacetate (*Figure 6—figure supplement 5B*, *Supplementary file 3*). The resulting $K_M$ of 7 mM and a $k_{cat}$ of 25.9 s$^{-1}$ (*Supplementary file 3*) indicated a sufficient turnover of serine and oxaloacetate into aspartate and hydroxypyruvate. To further verify that the strain is indeed using the amine from glycine or serine, respectively, we conducted a labeling experiment growing the glut-aux Δ*dadA* overexpressing BhcA and a WT control on M9 with $^{15}NH_4$ or M9 with $^{14}NH_4$ supplemented with 20 mM glycerol and 20 mM glycine, or 1.7 mM serine. Aspartate, proline, serine, alanine, and glycine were unlabeled in the glut-aux Δ*dadA* strain overexpressing BhcA, confirming that indeed all amines were derived only from the provided amino acid and not from free ammonium in the medium (*Figure 6—figure supplement 7*).

## Exploring alternative routes of ammonium assimilation

After testing and engineering the glut-aux strain to use amino acids as amine group donors, we aimed to explore alternative ammonium assimilation pathways, thus rewiring canonical ammonium assimilation via glutamate. For this, we used the ammonium assimilation-deficient glut-aux strain to test the

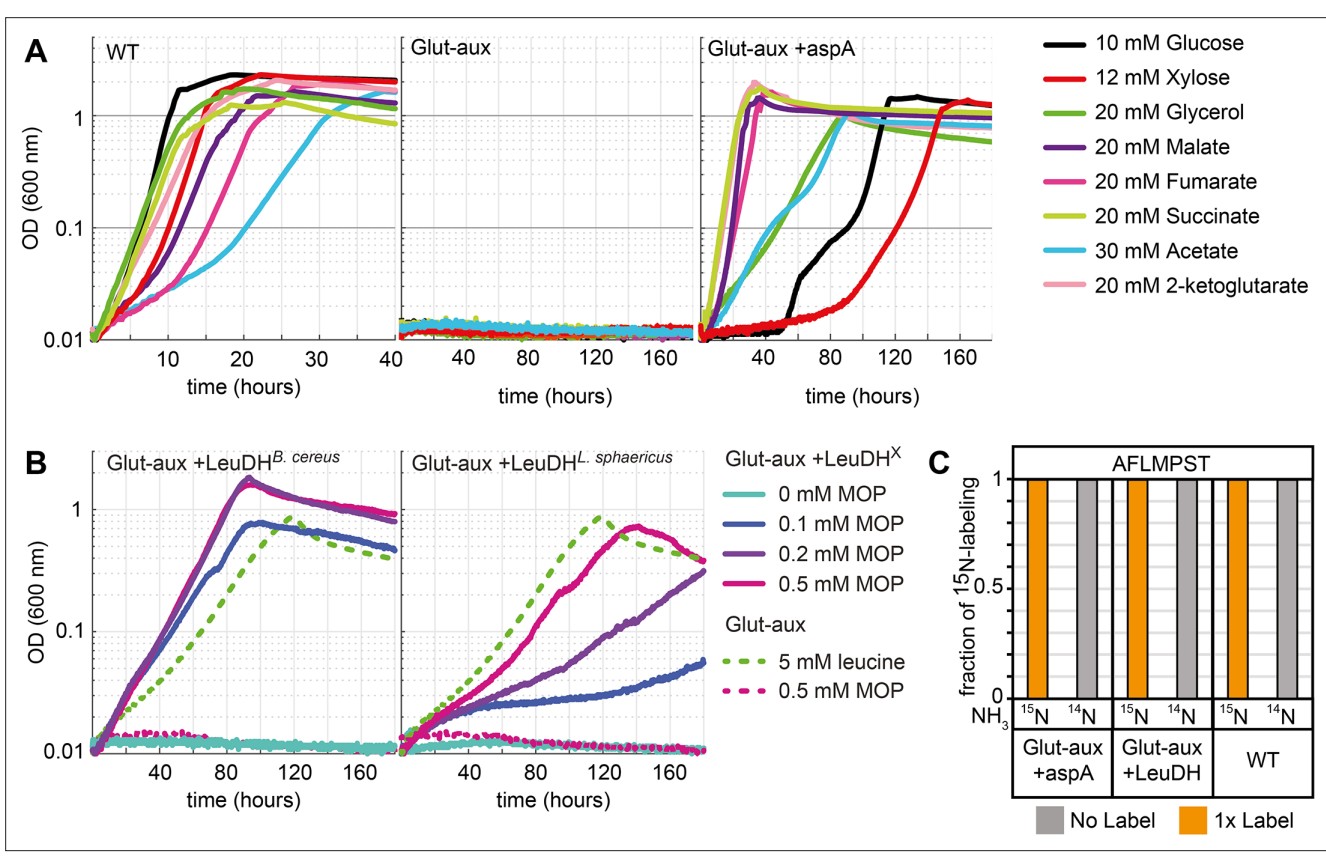

**Figure 7.** Aspartate ammonia lyase or leucine dehydrogenase (LeuDH) can replace glutamate-based ammonia assimilation to fix all ammonium for cell growth. (**A**) Growth of wildtype (WT), glutamate auxotrophic strain (glut-aux), and glut-aux + aspA on M9 with ammonium and indicated carbon sources. Overexpression of aspartate ammonia lyase allows the glut-aux to assimilate free ammonium via fumarate amination. (**B**) Growth of glut-aux and glut-aux + LeuDH from *Bacillus cereus* or *Lysinibacillus sphaericus* on M9 with ammonium, 20 mM glycerol as carbon source and the indicated additional substrates. Overexpression of LeuDH allows the glut-aux to assimilate free ammonium from the medium in presence of 4-methyl-2-oxopentanoate (MOP). (**C**) Representative dataset for nitrogen labeling results from glut-aux + AspA, glut-aux + LeuDH, and WT strains. Cells were grown either with $^{15}NH_4$ or $^{14}NH_4$ and 20 mM succinate (glut-aux + AspA), 20 mM glycerol + 0.5 mM MOP (glut-aux + LeuDH, from *B. cereus* or *L. sphaericus*), or 20 mM glycerol (WT). All strains were grown in M9 medium with ammonium. Data represents triplicate experiments with <5% variation.

enzymes aspartate ammonia lyase (AspA) and leucine dehydrogenase (LeuDH) for their activity to supply all cellular amines for growth. Both aspartate and leucine previously were amine sources for the glut-aux strain, but while aspartate allowed fastest growth of all amino acids tested, leucine supported only very slow growth (*Figure 2*, *Supplementary file 2*).

## Aspartate ammonia lyase

Since aspartate served as an efficient amine donor (*Figure 2*), we investigated whether it could also replace glutamate as the formation product of ammonium assimilation via the reverse activity of aspartate ammonia lyase (AspA; UniProt ID P0AC38, fumarate + $NH_4$ = aspartate), an *E. coli* native reaction. This enzyme canonically operates as part of the aspartate utilization/degradation pathway releasing ammonia. However, the thermodynamics of this reaction reveal a full reversibility with a $\Delta_r G'^m$ of 4 [kJ/mol] (in the aspartate-forming direction at a concentration of 1 mM for all reactants; http://equilibrator.weizmann.ac.il/). We hence overexpressed AspA in the glut-aux strain and analyzed its growth on a variety of carbon sources (*Figure 7A*). Here, growth of the strain is dependent on both the ammonium assimilation activity to form aspartate and the subsequent aminotransferase activity to rescue the glut-aux strain (via rescue mechanism II, *Figure 1*). Growth of the glut-aux strain over-expressing AspA (+AspA) was indeed observed on all tested carbon sources (*Figure 7A*), suggesting that aspartate and AspA can replace the canonical glutamate-based ammonium assimilation pathway. The glut-aux + AspA strain grew fastest on carbon sources closest to fumarate, the substrate of AspA (2-ketoglutarate, succinate, malate, fumarate). Other carbon sources (glucose, xylose, glycerol, acetate), which do not directly generate fumarate, also supported growth, but with much higher doubling times. Assimilation of free ammonium from the medium by the glut-aux + AspA strain was confirmed by a $^{15}N$ labeling experiment comparing proteinogenic amino acid labeling of the glut-aux + AspA strain with the WT (serving as a positive control) upon growth on 20 mM succinate with either $^{15}NH_4$ or $^{14}NH_4$. Here, the WT and the glut-aux + AspA strain, showed identical labeling (*Figure 7C*).

## Leucine dehydrogenase

In our initial experiments, we showed that leucine can serve as an amine group donor in the glut-aux strain (*Figure 2*, via rescue mechanism II). To test whether we can engineer the glut-aux strain to produce leucine for the concomitant transaminase reactions, we overexpressed leucine dehydrogenase (LeuDH), which catalyzes the reductive amination of 4-methyl-2-oxopentanoate (MOP) to leucine. LeuDH is absent in the repertoire of *E. coli's* enzymes, hence we separately overexpressed two LeuDH genes, one from *Bacillus cereus* (*Li et al., 2009*) and the other from *Lysinibacillus sphaericus* (*Lu et al., 2016*). In initial experiments, the glut-aux + LeuDH did not grow in minimal media with 20 mM glycerol. As MOP is not a central metabolite, its concentration might be limiting the activity of the enzyme. Consequently, we observed growth of the glut-aux strain only upon supplementation with catalytic amounts of MOP, which served as ammonium acceptor in the amination reaction catalyzed by LeuDH (*Figure 7B*). We note that the MOP concentrations supplemented to the medium were at least tenfold lower compared to the 5 mM leucine addition, which served as a positive control in the experiments. The glut-aux strain overexpressing *B. cereus* LeuDH growing on glycerol and 0.2 and 0.5 mM MOP reached twice the $OD_{600}$ obtained for the glut-aux strain growing on glycerol and 5 mM leucine. This yield difference indicates a recycling of MOP through alternating activities of LeuDH and transaminases, fixing ammonium and transferring amine groups to other ketoacids. Growth of the glut-aux strain overexpressing *L. sphaericus* LeuDH resulted in lower yields than achieved with *B. cereus* LeuDH overexpression, and also lower yields than seen for the glut-aux with 5 mM leucine, indicating lower enzyme efficiency under the tested conditions. To verify ammonium assimilation via LeuDHs, the $^{15}N$ labeling in proteinogenic amino acids was analyzed after growing the glut-aux + LeuDH strains with 20 mM glycerol, 0.5 mM MOP, and either with $^{15}NH_4$ or $^{14}NH_4$ (*Figure 7C*). The observed single label for the analyzed amino acids alanine, phenylalanine, methionine, proline, serine, and threonine once again confirms operation of the LeuDH's ammonium entry point (*Figure 7C*).

## Discussion

Our study provides a broad investigation of amino acid and ammonium metabolism in *E. coli*. To examine which amino acids can be used as an amine source to compensate for the absence of

canonical glutamate production, we created a glutamate auxotrophic (glut-aux) strain and tested its ability to grow on different amino acids (*Figure 2*, supportive summary in *Figure 2—figure supplement 1*). Our experiments revealed that the glut-aux did not grow on all tested amino acids. Here, we note that the substrate spectrum of the underlying amination network might be larger than identified by our experiments since for some amino acids we cannot rule out that their uptake limits their intracellular availability to the network. Specifically, this might be the case for valine, threonine, and β-alanine, which did not support growth of any strain. However, for some of these, previous studies identified the mechanisms preventing utilization as an amine source. For example, use of threonine as nitrogen source would require induction of threonine dehydrogenase by the presence of leucine (*Potter et al., 1977*). Growth in the presence of valine was shown to be inhibited due to valine-dependent inhibition of acetohydroxy acid synthase essential to isoleucine biosynthesis (*Tuite et al., 2005*). In contrast, other amino acids known to be suitable nitrogen sources for the wildtype and canonical transamination products of glutamate did not rescue growth of the glut-aux strain via the reverse reaction (i.e., alanine). For this amino acid, we identified alanine degradation as the counteracting pathway. The glut-aux strain was capable of growth on most amino acids (proline, aspartate, asparagine, methionine, leucine, isoleucine, valine, glutamate, glutamine, ornithine, tyrosine, phenylalanine, tryptophan, and histidine). As absence of nitrogen from the medium only affected growth of the glut-aux on ornithine, we concluded that nitrogen response-dependent regulation was of minor relevance for amino acid metabolism. We assumed that in order for the glut-aux strain to grow, amino acids could either (i) fully replace glutamate as an amine donor, (ii) donate their amine directly to 2-ketoglutarate, (iii) be converted into an intermediate donating an amine to 2-ketoglutarate to form glutamate, or (iv) be direct glutamate precursors. Given the inability of the selection strain to synthesize glutamate, intracellular glutamate levels are expected to be lower than in a wildtype strain under saturating nitrogen availability. Therefore, some of the identified rescue mechanisms might not be the universally favored metabolic options. However, defining the mechanisms of growth restoration in the glut-aux strain might give insights into strategies of adaptation to limiting ammonium availability. To investigate which reactions allowed amine utilization from amino acids, we deleted several genes responsible for the respective amino acid degradation pathway. While proline was directly converted to glutamate (rescue mechanism IV, *Figure 1*), utilization of histidine, tyrosine, phenylalanine, tryptophan, methionine, isoleucine, and leucine depended on transamination mediated by AspC, one of the three main transaminases in *E. coli* (*Gelfand and Steinberg, 1977*; rescue mechanism II, *Figure 1*). Three aspects of this finding were surprising to us. First, deletion of AspC alone was sufficient to abolish growth of the glut-aux strain on tyrosine, phenylalanine, isoleucine, and leucine. This was unexpected since the other two main transaminases TyrB and IlvE share cross-reactivities with these amino acids and indicates that in spite of their cross-reactivity they cannot fully replace AspC in vivo. Residual tyrosine and phenylalanine aminotransferase activity in *tyrB* and *ilvE* knockout strains in previous works (*Gelfand and Steinberg, 1977*) led to the hypothesis that AspC was mainly exhibiting these activities, which herewith is further supported. Second, deletion of AspC abolished growth on histidine, methionine, isoleucine, and leucine for which only low or no specific activity of AspC was reported in vitro (*Hayashi et al., 1993*; *Powell and Morrison, 1978*). Under physiological conditions, the substrate range of AspC thus seems to be broader than previously anticipated. Third, the glut-aux Δ*aspC* strain was unable to grow with tryptophan as an amine source. Although activity of AspC with tryptophan was demonstrated in vitro (*Hayashi et al., 1993*; *Powell and Morrison, 1978*), transamination is not a known tryptophan degradation pathway in *E. coli* (*Reitzer, 2014*). We hence conclude that in the glut-aux strain, tryptophan is the direct substrate of AspC to form (indole-3-yl)-pyruvate and glutamate as in some bacteria (e.g., *P. aeruginosa* [*Bortolotti et al., 2016*], *Clostridium sporogenes* [*O'Neil and DeMoss, 1968*]), protozoa (*Trichomonas vaginalis* [*Lowe and Rowe, 1985*]), or mammals (*Shrawder and Martinez-Carrion, 1972*). Grouping amino acids by expected rescue mechanism and rescue phenotype (*Figure 2—figure supplement 1*) revealed that the majority of amino acids we suggest to be used via direct transamination rescued the glut-aux strain, but were not a good amine source for the wildtype (*Figure 2*, *Figure 2—figure supplement 1*). Additionally, we found AspC to be essential for the growth of the glut-aux on all amino acids not supporting WT growth (*Figure 5*). Therefore, we suggest promiscuous transaminase activity here rescuing growth of the glut-aux. strain to be one mechanism for adaptation to lacking glutamate availability. Hence,

our study expands previous knowledge by several novel transamination routes in *E. coli,* which need to be considered in contexts of low amine or glutamate availability.

In addition to identifying natively present routes of amino acid metabolism, we demonstrated how amino acid metabolism can be rewired to allow utilization of new amine sources. Overexpression of alanine-2-ketoglutarate aminotransferase, together with the removal of an alanine sink, allowed usage of alanine as an amine source. By overexpressing transaminases that transfer amines from β-alanine to 2-ketoglutarate, we engineered usage of the non-proteinogenic amino acid β-alanine as amine source. Following the same logic, we expressed an aspartate-glyoxylate aminotransferase to select for its reverse activity with glycine or serine. Upon deletion of the D-amino acid dehydrogenase gene *dadA* as potential internal sink for glycine and serine, the engineered strain was able to use the amine from glycine to support growth. Surprisingly, we discovered that the strain did not use glycine directly, but converted it to serine, which was yet another substrate of the transaminase showing high activity for serine ($k_{cat}$ of 25.87 s$^{-1}$) with oxaloacetate as amine acceptor. In principle, the newly discovered serine-oxaloacetate transaminase activity can be coupled to formate assimilation via the reductive glycine pathway (*Kim et al., 2020*), which fixes ammonium by reverse activity of the glycine cleavage system. This allows ammonium transfer to make aspartate and convert serine to hydroxypyruvate instead of pyruvate (generated via serine deaminase activity in the reductive glycine pathway), hence saving ATP needed to convert pyruvate into PEP, essential for anaplerosis and gluconeogenesis. The usage of alanine, β-alanine, glycine, and serine as amine donors was engineered in the glut-aux strain by transaminase overexpression and (if necessary) removal of intracellular substrate sinks. Thus, we conclude that for these amino acids, transaminase availability was the main bottleneck rather than transport or regulatory issues. In addition to this finding, these examples demonstrate the glut-aux strain as an excellent selection platform to screen for any reaction that produces amino acids which can rescue growth. This potentially allows the discovery of new amine sources.

The function of glutamate as a universal amine transfer molecule originates in its role as canonical entry point of ammonium into metabolism (*Kumada et al., 1993*). We were interested in finding out whether the prominence of this ammonium assimilation mechanism can be explained by a significant advantage compared to other mechanisms of ammonium assimilation. To investigate this, we replaced canonical ammonium assimilation via glutamate biosynthesis with two alternative pathways. Both fumarate amination to aspartate and MOP amination to leucine reconstituted the ability of the glut-aux strain to assimilate free ammonium. Growth dependent on ammonium assimilation via AspA was fastest when a carbon source metabolically close to fumarate was provided. We note that assimilation of ammonium via aspartate ammonia lyase is energetically different from ammonium assimilation via glutamate. While a glutamate-dependent amination network requires NADPH consumption by glutamate dehydrogenase, an aspartate ammonia lyase-dependent network requires NADH for the reduction of oxaloacetate back to fumarate; oxaloacetate being the transamination product of *aspC*. NADH is energetically cheaper compared to NADPH as the recovery of the latter from NADH wastes some proton motive force through the membrane-bound transhydrogenase, and hence comes with the indirect cost of ATP synthesis (*Spaans et al., 2015*). Although ammonium assimilation via aspartate or leucine biosynthesis led to immediate growth of the glut-aux strain in growth-optimized laboratory conditions, these do not reflect natural conditions under which glutamate-based ammonium assimilation might have evolved and become prominent. Under ammonium-limiting conditions, GDH and GS are kinetically superior with a $K_M$ of 2 mM and 0.1 mM for ammonia, respectively (*Reitzer, 2014*), compared to AspA or LeuDH from *B. cereus* with $K_M$ values of 20 mM and 13 mM for ammonia, respectively (*Suzuki et al., 1973*; *Sanwal and Zink, 1961*). Additionally, the availability of two different systems (GDH and GS/GOGAT), which are each optimal for different growth conditions but generate the same molecule, is unique to glutamate-based ammonium assimilation. Ammonium assimilation as metabolic link between energy, carbon, and nitrogen metabolism requires a tight multilayer regulation of GDH and GS. Thus, immediate growth via alternative ammonium assimilation routes that are independent of native transcriptional regulation, as achieved via aspartate and leucine in our study, is nontrivial. On the other hand, the laboratory cultivation conditions that allow this immediate growth do not require the metabolic flexibility and control given by the interplay and regulation of GDH and GS. It is even likely that growth of the engineered strains under nonoptimal conditions would be impaired by a lack of flexible metabolic responses to varying nitrogen and energy availability. This might explain the conservation of glutamate-coupled ammonium assimilation as opposed to

alternative ammonium assimilation pathways. Furthermore, regulatory effects, for example, GS inhibition by several amino acids (e.g., tryptophan, histidine, alanine, glycine), might explain why some amino acids only supported slow growth of the glut-aux strain (*Hubbard and Stadtman, 1967*).

The possibility to modify ammonium assimilation and metabolism by metabolic engineering indicates that the underlying metabolic network is highly flexible, which is supported by highly promiscuous transaminases. However, transaminases are a limiting factor that defines the amination network, which implies the presence of different amination networks in different organisms that express different transaminases. That said, our study demonstrates how amino acid usage and even ammonium assimilation mechanisms are transferable by expressing heterologous genes. In this, we find that to allow such transferability between apparently different amination networks, the amine distribution network around the universally conserved core principle of glutamate-based amine assimilation must be highly flexible. Additionally, these engineering possibilities might have relevance for the metabolic engineering of synthetic pathways, for example, for growth-coupled selection (*Orsi et al., 2021*) or the production of certain amino acids or their derivatives. By engineering ammonium assimilation via non-glutamate-producing reactions, the cellular ammonium assimilation can be decoupled from glutamate biosynthesis for the production of nitrogenous compounds. Such decoupling follows the paradigm of modular design for bioengineering, which can support new endeavors for strain creation for biotechnological production processes. Here, the most obvious impact could be made on the

**Table 1.** Strains and plasmids used in this study.

| Name | Description/deletions | References |
|---|---|---|
| *E. coli strains* | | |
| DH5α | Cloning of overexpression constructs | |
| SIJ488 | WT, integrated $\lambda$ -red recombinase and flippase | *Jensen et al., 2015* |
| Glut-aux | Δ*gdhA* Δ*gltBD* | This study |
| Glut-aux Δ*ybdL* | Δ*gdhA* Δ*gltBD* Δ*ybdL* | This study |
| Glut-aux Δ*putA* | Δ*gdhA* Δ*gltBD* Δ*putA* | This study |
| Δ*putA* | Δ*putA* | This study |
| Glut-aux Δ*aspC* | Δ*gdhA* Δ*gltBD* Δ*aspC* | This study |
| Glut-aux Δ*dadX* | Δ*gdhA* Δ*gltBD* Δ*dadX* | This study |
| Glut-aux Δ*dadA* | Δ*gdhA* Δ*gltBD* Δ*dadA* | This study |
| Glut-aux Δ*gcv* | Δ*gdhA* Δ*gltBD* Δ*gcvTHP* | This study |
| Glut-aux Δ*gcv* Δ*dadA* | Δ*gdhA* Δ*gltBD* Δ*gcvTHP* Δ*dadA* | This study |
| JW0999 | KEIO Δ*putA* | *Baba et al., 2006* |
| *Plasmids* | | |
| ASS | Overexpression plasmid with p15A origin, streptomycin resistance, constitutive strong promoter | *Wenk et al., 2018* |
| ASS-aspA | ASS backbone for overexpression of *aspA* from *E. coli* | This study |
| ASS-alaA | ASS backbone for overexpression of *alaA* from *E. coli* | This study |
| ASS-alaC | ASS backbone for overexpression of *alaC* from *E. coli* | This study |
| ASS-leuDH[B. cereus] | ASS backbone for overexpression of leucine dehydrogenase from *Bacillus cereus* | This study |
| ASS-leuDH[L. sphaericus] | ASS backbone for overexpression of leucine dehydrogenase from *Lysinibacillus sphaericus* | This study |
| ASS-BAPAT | ASS backbone for overexpression of β-alanine-pyruvate-aminotransferase from *Pseudomonas aeruginosa* | This study |
| ASS-BAOAT | ASS backbone for overexpression of β-alanine-2-ketoglutarate-aminotransferase from *Saccharomyces kluyveri* | This study |
| ASS-BhcA | ASS backbone for overexpression of glycine-oxaloacetate aminotransferase from *Paracoccus denitrificans* | This study |

million-ton-scale industrial amino acid production (*Wendisch, 2020*). Engineering new ammonium entry points as well as extending the amination network could positively affect these processes. For example, such metabolic network optimization was previously demonstrated by increased arginine production yields in *Corynebacterium glutamicum* upon expression of *E. coli* AspA and GlnA (*Guo et al., 2017*).

Altogether, our research addressed three distinct and complementary aspects regarding amino acid and ammonium metabolism. We provided a comprehensive overview of the options and limitations of the cellular amination network. Then, we showed its flexibility when engineering the network towards the use of new amine sources. Finally, by engineering new ammonium entry points, we increased the potential design space for engineering ammonium assimilation and dissimilation.

## Materials and methods

### Strains

All *E. coli* strains used in this study are listed in *Table 1*. Strain SIJ488, which carries inducible recombinase and flippase genes (*Jensen et al., 2015*), was used as wildtype for the generation of deletions. Gene deletions were performed by $\lambda$-red recombineering or P1-transduction as described below.

### Gene deletion via P1 transduction

Deletions of *putA* and *dadX* were generated by P1 phage transduction (*Thomason et al., 2007*). Strains from the Keio collection carrying single gene deletions with a kanamycin-resistance gene (KmR) as selective marker were used as donor strains (*Baba et al., 2006*). Selection for strains that had acquired the desired deletion was performed by plating on appropriate antibiotics (kanamycin [Km]) and confirmed by determining the size of the respective genomic locus via PCR using DreamTaq polymerase (Thermo Scientific, Dreieich, Germany) and the respective KO-Ver primers (*Supplementary file 5*). Additionally, it was confirmed that no copy of the gene to be deleted was present anywhere in the genome by PCR using DreamTaq polymerase (Thermo Scientific) and internal primers binding inside of the coding sequence of the gene. To remove the selective marker, a fresh culture was grown to $OD_{600}$ ~ 0.2, followed by inducing flippase expression by adding 50 mM L-rhamnose and cultivating for ~4 hr at 30°C. Loss of the antibiotic resistance was confirmed by identifying individual colonies that only grew on LB in the absence of the respective antibiotic and by PCR of the genomic locus using the locus-specific KO-Ver primers.

### Gene deletion by recombineering

For gene deletion by recombineering, Km resistance cassettes were generated via PCR – 'KO' primers with 50 bp homologous arms are listed in *Supplementary file 5* – using the Km cassette from pKD4 (pKD4 was a gift from Barry L. Wanner [Addgene plasmid# 45605; http://n2t.net/addgene:45605; RRID:Addgene_45605 *Datsenko and Wanner, 2000*; *Baba et al., 2006*]), or in case of the *aspC* deletion the chloramphenicol (Cap) cassette from pKD3 (pKD3 was a gift from Barry L. Wanner [Addgene plasmid# 45604; http://n2t.net/addgene:45604; RRID:Addgene_45604]). To prepare cells for gene deletion, fresh cultures were inoculated in LB and the recombinase genes were induced by the addition of 15 mM L-arabinose at OD ~0.4–0.5. After incubation for 45 min at 37°C, cells were harvested and washed three times with ice-cold 10% glycerol (11,000 rpm, 30 s, 2°C). Approximately 300 ng of Km cassette PCR product was transformed via electroporation (1 mm cuvette, 1.8 kV, 25 μF, 200 Ω). After selection on Km, gene deletions were confirmed via PCR using 'KO-Ver' primers (*Supplementary file 5*). To remove the Km cassette, 50 mM L-rhamnose, which induces flippase gene expression, was added to an exponentially growing 2 mL LB culture at OD 0.5; induction time was ≥3 hr at 30°C. Colonies were screened for Km sensitivity, and removal of antibiotic resistance cassette was confirmed by PCR (using 'KO-Ver' primers).

### Plasmid construction

For overexpression, genes encoding for LeuDHs from *B. cereus* (LeuDH[B. cereus], P0A392) and *L. sphaericus* (LeuDH[L. sphaericus], B1HRW1), aspartate ammonia lyase from *E. coli* (AspA, P0AC38), β-alanine ketoglutarate transaminase from *S. kluyveri* (BAOAT, A5H0J5), β-alanine pyruvate transaminase from *P. aeruginosa* (BAPAT, Q9I700), and glycine-oxaloacetate aminotransferase from *P. denitrificans* (BhcA,

A1B8Z3) were synthesized after removal of restriction sites relevant for cloning (*Zelcbuch et al., 2013*) and codon adaptation to *E. coli*'s codon usage (*Grote et al., 2005*). Genes were synthesized by Twist Bioscience (San Francisco, CA). Alanine transaminase genes *alaA* and *alaC* were amplified from *E. coli*'s genome with high-fidelity Phusion Polymerase (Thermo Scientific) using primer pairs *alaA*-amp_fwd, *alaA*-amp_rvs and *alaC*-amp_fwd, *alaC*-amp_rvs, respectively (*Supplementary file 5*).

Cloning was carried out in *E. coli* DH5α. All genes were cloned via *Mph1103*I and *Xho*I into pNivC vector downstream of ribosome binding site 'C' (AAGTTAAGAGGCAAGA) (*Zelcbuch et al., 2013*). Restriction enzymes *EcoR*I and *Pst*I (FastDigest, Thermo Scientific) were used to transfer the genes into the expression vector pZ-ASS (p15A origin, streptomycin resistance, strong promoter) (*Braatsch et al., 2008*). Constructed vectors were confirmed by Sanger sequencing (LGC Genomics, Berlin, DE). Geneious 8 (Biomatters, New Zealand) was used for in silico cloning and sequence analysis.

The plasmid for the expression of the *dadA* gene was retrieved from the ASKA collection (*Kitagawa et al., 2005*). The plasmids for the expression of the *bhcA* and *ghrA* genes were described in previous publications (*Miller et al., 2020*; *Schada von Borzyskowski et al., 2019*).

## Media and growth experiments

LB medium (1% NaCl, 0.5% yeast extract, 1% tryptone) was used for cloning, generation of deletion strains, and strain maintenance. When appropriate, Km (25 µg/mL), ampicillin (100 µg/mL), streptomycin (100 µg/mL), or chloramphenicol (30 µg/mL) were used but omitted for growth experiments. Growth experiments were carried out in standard M9 minimal media (50 mM $Na_2HPO_4$, 20 mM $KH_2PO_4$, 1 mM NaCl, 20 mM $NH_4Cl$, 2 mM $MgSO_4$ and 100 µM $CaCl_2$, 134 µM EDTA, 13 µM $FeCl_3\cdot6H_2O$, 6.2 µM $ZnCl_2$, 0.76 µM $CuCl_2\cdot2H_2O$, 0.42 µM $CoCl_2\cdot2H_2O$, 1.62 µM $H_3BO_3$, 0.081 µM $MnCl_2\cdot4H_2O$) or in M9 media lacking $NH_4Cl$. Carbon sources were used as indicated in the text. Unless otherwise specified, L-isomers of amino acids were used if extant. For growth experiments, overnight cultures were incubated in 4 mL M9 medium containing 20 mM glycerol supplemented with 5 mM aspartate. Cultures were harvested ($6000\times g$, 3 min) and washed three times in M9 medium (w/ or w/o $NH_4Cl$) to remove residual carbon and $NH_4Cl$ sources. Washed cells were used to inoculate growth experiments to an optical density ($OD_{600}$) of 0.01 in 96-well microtiter plates (Nunclon Delta Surface, Thermo Scientific) at 37°C. Each well contained 150 µL of culture and 50 µL mineral oil (Sigma-Aldrich) to avoid evaporation while allowing gas exchange. Growth was monitored in technical triplicates if not stated otherwise at 37°C in BioTek Epoch 2 Microtiterplate reader (BioTek, Bad Friedrichshall, Germany) by absorbance measurements ($OD_{600}$) of each well every ~10 min with intermittent orbital and linear shaking. Blank measurements were subtracted and $OD_{600}$ measurements were converted to cuvette $OD_{600}$ values by multiplying with a factor of 4.35 as previously established empirically for the instruments. Growth curves were plotted in MATLAB and represent averages of measurements of technical replicates.

## Isolation and sequence analysis of glut-aux ASS-*alaA* mutants

The glut-aux strain + *alaA* was inoculated to $OD_{600}$ of 0.02 in tube cultures of 4 mL M9 + 20 mM glycerol + 5 mM alanine. Cell growth was monitored during prolonged incubation at 37°C for up to 14 days. Within that time several cultures started to grow and reached an OD above 1.0. Cells were streaked out on LB plates with streptomycin (to maintain the pZ-ASS-*alaA* plasmid) by dilution streak to generate single colonies. Isolates were inoculated into tube cultures of 4 mL M9 + 20 mM glycerol + 5 mM alanine, and the ones that immediately grew were used in genome sequence analysis. Genomic DNA was extracted using the Macherey-Nagel NucleoSpin Microbial DNA purification Kit (Macherey-Nagel, Düren, Germany) from $2 \times 10^9$ cells of an overnight culture in LB medium supplied with streptomycin and chloramphenicol (to maintain pZ-ASS-*alaA* plasmid). Construction of (microbial short insert libraries) PCR-free libraries for single-nucleotide variant detection and generation of 150 bp paired-end reads on an Illumina HiSeq 3000 platform were performed by Novogene (Cambridge, UK). Reads were mapped to the reference genome of *E. coli* MG1655 (GenBank accession no. U00096.3) using Breseq (Barrick Lab, TX) (*Deatherage and Barrick, 2014*). Using algorithms supplied by the software package, we identified single-nucleotide variants (with >50% prevalence in all mapped reads) and searched for regions with coverage deviating more than 2 SDs from the global median coverage.

## $^{15}$N isotopic labeling of proteinogenic amino acids

To elucidate the origin of the nitrogen in amino acids, we used $^{15}$N isotope-tracing experiments. Proteinogenic amino acids were analyzed after cell growth in M9 containing $^{15}$NH$_4$Cl (Sigma-Aldrich, Germany). Cells (1 mL of OD$_{600}$ 1) were harvested by centrifugation (6000×g) after reaching stationary phase and washed in H$_2$O. Proteins were hydrolyzed in 6 N HCl, at 95°C for 24 hr (*You et al., 2012*). HCl was removed by evaporation under an air stream at 95°C. Samples were then resuspended in 1 mL H$_2$O, insoluble compounds were removed by centrifugation (10 min, 16,000×g), and supernatants were used for analysis. Amino acid masses were analyzed by UPLC-ESI-MS as described previously (*Giavalisco et al., 2011*) with a Waters Acquity UPLC system (Waters) using a HSS T3 C18 reversed-phase column (100 mm × 2.1 mm, 1.8 µm; Waters). The mobile phases were 0.1% formic acid in H$_2$O (A) and 0.1% formic acid in acetonitrile (B). The flow rate was 0.4 mL/min with a gradient of 0–1 min – 99% A; 1–5 min – linear gradient from 99% A to 82%; 5–6 min – linear gradient from 82% A to 1% A; 6–8 min – kept at 1% A; 8–8.5 min – linear gradient to 99% A; 8.5–11 min – re-equilibrate. Mass spectra were acquired using an Exactive mass spectrometer (Thermo Scientific) in positive ionization mode, with a scan range of 50.0–300.0 m/z. The spectra were recorded during the first 5 min of the LC gradients. Data analysis was performed using Xcalibur (Thermo Scientific). Amino acid standards (Sigma-Aldrich) were analyzed for the determination of the retention times under the same conditions.

## Protein purification

To produce DadA and GhrA proteins for in vitro characterization, *E. coli* BL21 cells were transformed with the plasmid containing the respective gene. The cells were then grown on LB agar plates containing 50 µg/mL Km or 100 µg/mL ampicillin at 37°C overnight. Grown cells were used to inoculate a liter of selective terrific broth (TB). The expression cultures were grown overnight at 25°C in a shaking incubator. Their biomass was collected by centrifugation at 6000×g for 15 min at room temperature. The cells were resuspended in twice their volume of buffer A (50 mM HEPES/KOH, pH 7.8, 500 mM NaCl). The cells were lysed with a Microfluidizer (LM-10 H10Z, Microfludics, Westwood, USA) at 16.000 PSI for three passes on ice and the lysate was cleared by ultracentrifugation at 100,000×g for 45 min at 4°C and subsequently filtered through a 0.45 µm PTFE filter. The filtered lysate was loaded onto a 1 mL HisTrap FF (GE Healthcare, Freiburg, Germany) and unbound protein was removed with 20 column volumes of 15% buffer B (50 mM HEPES/KOH, pH 7.8, 500 mM NaCl, 500 mM imidazole) in buffer A. The protein was then eluted in 100% buffer B. The protein was desalted with a HiTrap 5 ml Desalting column (GE Healthcare) and a desalting buffer (50 mM HEPES/KOH, 50 mM NaCl, 20% [v/v] glycerol). Protein concentrations were determined by the protein's theoretical extinction coefficient and their absorbance at 280 nm. BhcA was expressed and purified as described previously (*Schada von Borzyskowski et al., 2019*).

## Measurement of enzyme activity

The activities of all tested enzymes were measured with a Cary 60 UV-Vis spectrophotometer (Agilent Technologies GmbH, Waldbronn, Germany) at 37°C. To test the transaminase activity of BhcA with serine, the absorbance at 365 nm of the reaction mix (50 mM HEPES/KOH, pH 7.5, 0.7 mM NADPH, 7 mM OAA, 30 µg GhrA, and 2.5 µg BhcA; modified based on *Schada von Borzyskowski et al., 2019*) was tracked over time. The reaction was started by adding varying concentrations of serine to the mixture, and the resulting slope in absorbance decrease was measured.

To test the activity of BhcA for oxaloacetate, another reaction mix (50 mM HEPES/KOH, pH 7.5, 0.7 mM NADPH, 100 mM serine, 30 µg GhrA, and 2.5 µg BhcA) was prepared. This time the reaction was started by adding oxaloacetate in varying concentrations to the mix. Activities of DadA were measured following the absorbance of dichlorophenolindophenole (DCPIP) at 600 nm. To measure activity with D-alanine, the reaction mix (200 mM HEPES/KOH, pH 7.5, 0.1 mM DCPIP, 1.5 mM PES, 10 mM KCN, 60 µg DadA) was started with varying concentrations of D-alanine. To measure the same reaction for glycine, the amount of DadA was increased to 280 µg, and the reaction was started with glycine.

## Acknowledgements

This work was funded by the Max Planck Society. SNL acknowledges support from the BMBF grant ForceYield (031B0825B). TJE acknowledges additional support from the German Research Foundation (SFB987 'Microbial diversity in environmental signal response'). The authors thank Beau Dronsella and Enrico Orsi for critical reading, corrections and discussions on the manuscript.

## Additional information

### Funding

| Funder | Grant reference number | Author |
|---|---|---|
| Bundesministerium für Bildung und Forschung | 031B0825B | Steffen N Lindner |
| Deutsche Forschungsgemeinschaft | SFB987 | Tobias J Erb |
| Max Planck Institute of Molecular Plant Physiology | open access funding | Helena Schulz-Mirbach Alexandra Müller Tong Wu Selçuk Aslan Arren Bar-Even Steffen N Lindner |
| Max Planck Institute for Terrestrial Microbiology | open access funding | Pascal Pfister |

The funders had no role in study design, data collection and interpretation, or the decision to submit the work for publication.

### Author contributions

Helena Schulz-Mirbach, Investigation, Writing – original draft, Writing – review and editing; Alexandra Müller, Tong Wu, Pascal Pfister, Investigation; Selçuk Aslan, Supervision; Lennart Schada von Borzyskowski, Conceptualization, Supervision, Writing – review and editing; Tobias J Erb, Supervision, Writing – review and editing; Arren Bar-Even, Conceptualization, Supervision, Writing – original draft; Steffen N Lindner, Conceptualization, Supervision, Writing – original draft, Writing – review and editing

### Author ORCIDs

Helena Schulz-Mirbach http://orcid.org/0000-0002-5376-9185
Arren Bar-Even http://orcid.org/0000-0002-1039-4328
Steffen N Lindner http://orcid.org/0000-0003-3226-3043

### Decision letter and Author response

Decision letter https://doi.org/10.7554/eLife.77492.sa1
Author response https://doi.org/10.7554/eLife.77492.sa2

## Additional files

### Supplementary files

• Supplementary file 1. Doubling time of glutamate auxotrophic (glut-aux) and wildtype on tested amino acids. Strains were grown over 170 hr in minimal medium with ammonium (glut-aux) and without ammonium (glut-aux, wildtype). 20 mM glycerol was used as the carbon source and 5 mM amino acid as nitrogen source. Maximal doubling time in a window of 6 hr was calculated using MATLAB and is indicated with standard error. An $OD_{600}$ of 0.1 was the threshold for defining growth (above $OD_{600}$ of 0.1) or no growth.

• Supplementary file 2. Mutations found in glutamate auxotrophic (glut-aux) + alaA mutants 1 and 2. After sequencing the genomes of two independently isolated glut-aux + alaA mutants and the glut-aux + alaA parent, the results were mapped against the *E. coli* MG1655 reference genome

(GenBank accession no. U00096.3) using Breseq. Mutations occurring in the mutants and not the parent are listed.

• Supplementary file 3. Kinetics of DadA with D-alanine and glycine. Enzyme activity was determined in a dichlorophenolindophenole (DCPIP)-coupled assay with various substrate concentrations and 60 μg protein for measurements with D-alanine and 280 μg protein for measurements with glycine. Data are mean ± SE.

• Supplementary file 4. Kinetics of BhcA with L-serine and oxaloacetate. Enzyme activity was determined in an assay coupling BhcA-dependent L-serine transamination to NADPH-dependent hydroxypyruvate reduction by GhrA with varying concentrations of L-serine or oxaloacetate for the respective activity measurements. Data are mean ± SE.

• Supplementary file 5. Oligonucleotide primers used. 'KO' primers were used to amplify the knockout kanamycin (Km) cassette from pKD4 with 50 bp gene-specific upstream and downstream sequences. 'KO-Ver' primers (knockout verification) were used to verify gene replacement by Km resistance cassette and cassette removal by flippase. External and internal primers were used to verify successful removal of the gene from the genome.

• Transparent reporting form

### Data availability

sequencing data has been deposited at Dryad.

The following dataset was generated:

| Author(s) | Year | Dataset title | Dataset URL | Database and Identifier |
|---|---|---|---|---|
| Schulz-Mirbach H, Müller A, Wu T, Pfister P, Aslan S, Schada von Borzyskowski L, Erb T, Bar-Even A, Lindner SN | 2022 | Data from: Flexibility of the cellular amination network in *Escherichia coli* | https://dx.doi.org/10.5061/dryad.mcvdnck2s | Dryad Digital Repository, 10.5061/dryad.mcvdnck2s |

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

## Appendix 1

### Sequences of synthetic genes

aspA
ATGCATCATCACCATCACCACTCTAACAACATCCGTATCGAAGAAGACCTGCTGGGTACCCGTG
AAGTTCCGGCTGACGCTTACTACGGTGTTCACACCCTGCGTGCTATCGAAAACTTCTACATCTC
TAACAACAAAATCTCTGACATCCCGGAATTTGTTCGTGGTATGGTTATGGTTAAAAAAGCTGCT
GCTATGGCTAACAAAGAACTGCAAACCATCCCGAAATCTGTTGCTAACGCTATCATCGCTGCTT
GCGACGAAGTTCTGAACAACGGTAAATGTATGGACCAGTTCCCGGTTGACGTTTACCAGGGTGG
TGCTGGTACCTCTGTTAACATGAACACCAACGAAGTTCTGGCTAACATCGGTCTGGAACTGATG
GGTCACCAGAAAGGTGAATACCAGTACCTGAACCCGAACGACCACGTTAACAAATGCCAGTCTA
CCAACGACGCTTACCCGACCGGTTTCCGTATCGCTGTTTACTCTTCTCTGATCAAACTGGTTGA
CGCTATCAACCAGCTGCGTGAAGGTTTCGAACGTAAAGCTGTTGAATTTCAGGACATCCTGAAA
ATGGGTCGTACCCAGCTGCAAGACGCTGTTCCGATGACCCTGGGTCAGGAATTTCGTGCTTTCT
CTATCCTGCTGAAAGAAGAAGTTAAAAACATCCAGCGTACCGCTGAACTGCTGCTGGAAGTTAA
CCTGGGTGCTACCGCTATCGGTACCGGTCTGAACACCCCGAAAGAATACTCTCCGCTGGCTGTT
AAAAAACTGGCTGAAGTTACCGGTTTCCCGTGCGTTCCGGCTGAAGACCTGATCGAAGCTACCT
CTGACTGCGGTGCTTACGTTATGGTTCACGGTGCTCTGAAACGTCTGGCTGTTAAAATGTCTAA
AATCTGCAACGACCTGCGTCTGCTGTCTTCTGGTCCGCGTGCTGGTCTGAACGAAATCAACCTG
CCGGAACTGCAAGCTGGTTCTTCTATCATGCCGGCTAAAGTTAACCCGGTTGTTCCGGAAGTTG
TTAACCAGGTTTGCTTCAAAGTTATCGGTAACGACACCACCGTTACCATGGCTGCTGAAGCTGG
TCAGCTGCAACTGAACGTTATGGAACCGGTTATCGGTCAGGCTATGTTCGAATCTGTTCACATC
CTGACCAACGCTTGCTACAACCTGCTGGAAAAATGTATCAACGGTATCACCGCTAACAAAGAAG
TTTGCGAAGGTTACGTTTACAACTCTATCGGTATCGTTACCTACCTGAACCCGTTCATCGGTCA
CCACAACGGTGACATCGTTGGTAAAATCTGCGCTGAAACCGGTAAATCTGTTCGTGAAGTTGTT
CTGGAACGTGGTCTGCTGACCGAAGCTGAACTGGACGACATCTTCTCTGTTCAGAACCTGATGC
ACCCGGCTTACAAAGCTAAACGTTACACCGACGAATCTGAACAGTAATCTAGAGCTAGCG

BaOAT
ATGCATCATCACCATCACCACCCGTCTTACTCTGTTGCTGAACTGTACTACCCGGACGAACCGA
CCGAACCGAAAATCTCTACCTCTTCTTACCCGGGTCCGAAAGCTAAACAGGAACTGGAAAAACT
GTCTAACGTTTTCGACACCCGTGCTGCTTACCTGCTGGCTGACTACTACAAATCTCGTGGTAAC
TACATCGTTGACCAGGACGGTAACGTTCTGCTGGACGTTTACGCTCAGATCTCTTCTATCGCTC
TGGGTTACAACAACCCGGAAATCCTGAAAGTTGCTAAATCTGACGCTATGTCTGTTGCTCTGGC
TAACCGTCCGGCTCTGGCTTGCTTCCCGTCTAACGACTACGGTCAGCTGCTGGAAGACGGTCTG
CTGAAAGCTGCTCCGCAGGGTCAGGACAAAATCTGGACCGCTCTGTCTGGTTCTGACGCTAACG
AAACCGCTTTCAAAGCTTGCTTCATGTACCAGGCTGCTAAAAAACGTAACGGTCGTTCTTTCTC
TACCGAAGAACTGGAATCTGTTATGGACAACCAGCTGCCGGGTACCTCTGAAATGGTTATCTGC
TCTTTCGAAAAAGGTTTCCACGGTCGTCTGTTCGGTTCTCTGTCTACCACCCGTTCTAAACCGA
TCCACAAACTGGACATCCCGGCTTTCAACTGGCCGAAAGCTCCGTTCCCGGACCTGAAATACCC
GCTGGAAGAAACAAAGAAGCTAACAAAGCTGAAGAATCTTCTTGCATCGAAAAATTCTCTCAG
ATCGTTCAGGAATGGCAGGGTAAAATCGCTGCTGTTATCATCGAACCGATCCAGTCTGAAGGTG
GTGACAACCACGCTTCTTCTGACTTCTTCCAGAAACTGCGTGAAATCACCATCGAAAACGGTAT
CCTGATGATCGTTGACGAAGTTCAGACCGGTGTTGGTGCTACCGGTAAATGTGGGCTCACGAA
CACTGGAACCTGTCTAACCCGCCGGACCTGGTTACCTTCTCTAAAAAATTCCAGGCTGCTGGTT
TCTACTACCACGACCCGAAACTGCAACCGGACCAGCCGTTCCGTCAGTTCAACACCTGGTGCGG
TGACCCGTCTAAAGCTCTGATCGCTAAAGTTATCTACGAAGAAATCGTTAAACACGACCTGGTT
ACCCGTACCGCTGAAGTTGGTAACTACCTGTTCAACCGTCTGGAAAAACTGTTCGAAGGTAAA
ACTACATCCAGAACCTGCGTGGTAAAGGTCAGGGTACCTACATCGCTTTCGACTTCGGTACCTC
TTCTGAACGTGACTCTTTCCTGTCTCGTCTGCGTTGCAACGGTGCTAACGTTGCTGGTTGCGGT
GACTCTGCTGTTCGTCTGCGTCCGTCTCTGACCTTCGAAGAAAACACGCTGACGTTCTGGTTT
CTATCTTCGACAAACCCTGCGTCAGCTGTACGGTTAA

BaPAT
ATGCATCATCACCATCACCACAACCAGCCGCTGAACGTTGCTCCGCCGGTTTCTTCTGAACTGA
ACCTGCGTGCTCACTGGATGCCGTTCTCTGCTAACCGTAACTTCCAGAAGACCCGCGTATCAT
CGTTGCTGCTGAAGGTTCTTGGCTGACCGACGACAAAGGTCGTAAAGTTTACGACTCTCTGTCT
GGTCTGTGGACCTGCGGTGCTGGTCACTCTCGTAAAGAAATCCAGGAAGCTGTTGCTCGTCAGC

TGGGTACCCTGGACTACTCTCCGGGTTTCCAGTACGGTCACCCGCTGTCTTTCCAGCTGGCTGA
AAAAATCGCTGGTCTGCTGCCGGGTGAACTGAACCACGTTTTCTTCACCGGTTCTGGTTCTGAA
TGCGCTGACACCTCTATCAAAATGGCTCGTGCTTACTGGCGTCTGAAAGGTCAGCCGCAGAAAA
CCAAACTGATCGGTCGTGCTCGTGGTTACCACGGTGTTAACGTTGCTGGTACCTCTCTGGGTGG
TATCGGTGGTAACCGTAAAATGTTCGGTCAGCTGATGGACGTTGACCACCTGCCGCACACCCTG
CAACCGGGTATGGCTTTCACCCGTGGTATGGCTCAGACCGGTGGTGTTGAACTGGCTAACGAAC
TGCTGAAACTGATCGAACTGCACGACGCTTCTAACATCGCTGCTGTTATCGTTGAACCGATGTC
TGGTTCTGCTGGTGTTCTGGTTCCGCCGGTTGGTTACCTGCAACGTCTGCGTGAAATCTGCGAC
CAGCACAACATCCTGCTGATCTTCGACGAAGTTATCACCGCTTTCGGTCGTCTGGGTACCTACT
CTGGTGCTGAATACTTCGGTGTTACCCCGGACCTGATGAACGTTGCTAAACAGGTTACCAACGG
TGCTGTTCCGATGGGTGCTGTTATCGCTTCTTCTGAAATCTACGACACCTTCATGAACCAGGCT
CTGCCGGAACACGCTGTTGAATTTTCTCACGGTTACACCTACTCTGCTCACCCGGTTGCTTGCG
CTGCTGGTCTGGCTGCTCTGGACATCCTGGCTCGTGACAACCTGGTTCAGCAGTCTGCTGAACT
GGCTCCGCACTTCGAAAAAGGTCTGCACGGTCTGCAAGGTGCTAAAAACGTTATCGACATCCGT
AACTGCGGTCTGGCTGGTGCTATCCAGATCGCTCCGCGTGACGGTGACCCGACCGTTCGTCCGT
TCGAAGCTGGTATGAAACTGTGGCAGCAGGGTTTCTACGTTCGTTTCGGTGGTGACACCCTGCA
ATTCGGTCCGACCTTCAACGCTCGTCCGGAAGAACTGGACCGTCTGTTCGACGCTGTTGGTGAA
GCTCTGAACGGTATCGCTTAATCTAGAGCTAGCG

## LeuDH<sup>B. cereus</sup>

ATGCATCATCACCATCACCACACCCTGGAAATCTTCGAATACCTGGAAAAATACGACTACGAAC
AGGTTGTTTTCTGCCAGGACAAAGAATCTGGTCTGAAAGCTATCATCGCTATCCACGACACCAC
CCTGGGTCCGGCTCTGGGTGGTACCCGTATGTGGACCTACGACTCTGAAGAAGCTGCTATCGAA
GACGCTCTGCGTCTGGCTAAAGGTATGACCTACAAAAACGCTGCTGCTGGTCTGAACCTGGGTG
GTGCTAAAACCGTTATCATCGGTGACCCGCGTAAAGACAAATCTGAAGCTATGTTCCGTGCTCT
GGGTCGTTACATCCAGGGTCTGAACGGTCGTTACATCACCGCTGAAGACGTTGGTACCACCGTT
GACGACATGGACATCATCCACGAAGAAACCGACTTCGTTACCGGTATCTCTCCGTCTTTCGGTT
CTTCTGGTAACCCGTCTCCGGTTACCGCTTACGGTGTTTACCGTGGTATGAAAGCTGCTGCTAA
AGAAGCTTTCGGTACCGACAACCTGGAAGGTAAAGTTATCGCTGTTCAGGGTGTTGGTAACGTT
GCTTACCACCTGTGCAAACACCTGCACGCTGAAGGTGCTAAACTGATCGTTACCGACATCAACA
AAGAAGCTGTTCAGCGTGCTGTTGAAGAATTTGGTGCTTCTGCTGTTGAACCGAACGAAATCTA
CGGTGTTGAATGCGACATCTACGCTCCGTGCGCTCTGGGTGCTACCGTTAACGACGAAACCATC
CCGCAGCTGAAAGCTAAAGTTATCGCTGGTTCTGCTAACAACCAGCTGAAAGAAGACCGTCACG
GTGACATCATCCACGAAATGGGTATCGTTTACGCTCCGGACTACGTTATCAACGCTGGTGGTGT
TATCAACGTTGCTGACGAACTGTACGGTTACAACCGTGAACGTGCTCTGAAACGTGTTGAATCT
ATCTACGACACCATCGCTAAAGTTATCGAAATCTCTAAACGTGACGGTATCGCTACCTACGTTG
CTGCTGACCGTCTGGCTGAAGAACGTATCGCTTCTCTGAAAAACTCTCGTTCTACCTACCTGCG
TAACGGTCACGACATCATCTCTCGTCGTTAA

## LeuDH<sup>L. sphaericus</sup>

ATGCATCATCACCATCACCACGAAATCTTCAAATACATGGAAAAATACGACTACGAACAGCTGG
TTTTCTGCCAGGACGAAGCTTCTGGTCTGAAAGCTGTTATCGCTATCCACGACACCACCCTGGG
TCCGGCTCTGGGTGGTGCTCGTATGTGGACCTACGCTTCTGAAGAAACGCTGTTGAAGACGCT
CTGCGTCTGGCTCGTGGTATGACCTACAAAAACGCTGCTGCTGGTCTGAACCTGGGTGGTGGTA
AAACCGTTATCATCGGTGACCCGTTCAAAGACAAAAACGAAGAAATGTTCCGTGCTCTGGGTCG
TTTCATCCAGGGTCTGAACGGTCGTTACATCACCGCTGAAGACGTTGGTACCACCGTTACCGAC
ATGGACCTGATCCACGAAGAAACCGACTACGTTACCGGTATCTCTCCGGCTTTCGGTTCTTCTG
GTAACCCGTCTCCGGTTACCGCTTACGGTGTTTACCGTGGTATGAAAGCTGCTGCTAAAGAAGC
TTTCGGTTCTGAATCTCTGGAAGGTCTGAAAATCTCTGTTCAGGGTCTGGGTAACGTTGCTTAC
AAACTGTGCGAATACCTGCACAACGAAGGTGCTAAACTGGTTGTTACCGACATCAACCAGGCTG
CTATCGACCGTGTTGTTAACGACTTCGACGCTATCGCTGTTGCTCCGGACGAAATCTACGCTCA
GGAAGTTGACATCTTCTCTCCGTGCGCTCTGGGTGCTATCCTGAACGACGAAACCATCCCGCAG
CTGAAAGCTAAAGTTATCGCTGGTTCTGCTAACAACCAGCTGAAAGACTCTCGTCACGGTGACT
TCCTGCACGAACTGGGTATCGTTTACGCTCCGGACTACGTTATCAACGCTGGTGGTGTTATCAA
CGTTGCTGACGAACTGTACGGTTACAACCGTGAACGTGCTCTGAAACGTGTTGACGGTATCTAC
GACTCTATCGAAAAAATCTTCGCTATCTCTAAACGTGACGGTATCCCGACCTACGTTGCTGCTA
ACCGTCTGGCTGAAGAACGTATCGCTCGTGTTGCTAAATCTCGTTCTCAGTTCCTGAAAAACGA
AAAAAACATCCTGCACGGTCGTTAA

BhcA

ATGCATCATCACCATCACCACACCTCTCAGAACCCGATCTTCATCCCGGGTCCGACCAACATCC
CGGAAGAAATGCGTAAAGCTGTTGACATGCCGACCATCGACCACCGTTCTCCGGTTTTCGGTCG
TATGCTGCACCCGGCTCTGGAAGGTGTTAAAAAAGTTCTGAAAACCACCCAGGCTCAGGTTTTC
CTGTTCCCGTCTACCGGTACCGGTGGTTGGGAAACCGCTATCACCAACACCCTGTCTCCGGGGTG
ACAAAGTTCTGGCTGCTCGTAACGGTATGTTCTCTCACCGTTGGATCGACATGTGCCAGCGTCA
CGGTCTGGACGTTACCTTCGTTGAAACCCCGTGGGGTGAAGGTGTTCCGGCTGACCGTTTCGAA
GAAATCCTGACCGCTGACAAAGGTCACGAAATCCGTGTTGTTCTGGCTACCCACAACGAAACCG
CTACCGGTGTTAAATCTGACATCGCTGCTGTTCGTCGTGCTCTGGACGCTGCTAAACACCCGGC
TCTGCTGTTCGTTGACGGTGTTTCTTCTATCGGTTCTATGGACTTCCGTATGGACGAATGGGGT
GTTGACATCGCTGTTACCGGTTCTCAGAAAGGTTTCATGCTGCCGCCGGGTCTGGCTATCGTTG
GTTTCTCTCCGAAAGCTATGGAAGCTGTTGAAACCGCTCGTCTGCCGCGTACCTTCTTCGACAT
CCGTGACATGGCTACCGGTTACGCTCGTAACGGTTACCCGTACACCCCGCCGGTTGGTCTGATC
AACGGTCTGAACGCTTCTTGCGAACGTATCCTGGCTGAAGGTCTGGAAAACGTTTTCGCTCGTC
ACCACCGTATCGCTTCTGGTGTTCGTGCTGCTGTTGACGCTTGGGGTCTGAAACTGTGCGCTGT
TCGTCCGGAACTGTACTCTGACTCTGTTTCTGCTATCCGTGTTCCGGAAGGTTTCGACGCTAAC
CTGATCGTTTCTCACGCTCTGGAAACCTACGACATGGCTTTCGGTACCGGTCTGGGTCAGGTTG
CTGGTAAAGTTTTCCGTATCGGTCACCTGGGTTCTCTGACCGACGCTATGGCTCTGTCTGGTAT
CGCTACCGCTGAAATGGTTATGGCTGACCTGGGTCTGCCGATCCAGCTGGGTTCTGGTGTTGCT
GCTGCTCAGGAACACTACCGTCAGACCACCGCTGCTGCTCAGAAAAAAGCTGCTTAATCTAGAG
CTAGC

