## [Editor Report]

This article investigates the non-canonical ammonium metabolism in *E. coli*, exploring possible ways by which *E. coli* is able to assimilate nitrogen. The work reveals that some amination reactions can be reversed and redirected with strain engineering, thus introducing the novel concept of a cellular amination network (reversible nitrogen transfer). In addition to furthering a fundamental understanding of metabolism, this work provides a modular platform for the metabolic engineering of ammonium assimilation and dissimilation, which has potential for the industrial bioproduction of nitrogenous compounds.

---

## [Decision Letter]

**Decision letter after peer review:**

Thank you for submitting your article "On the flexibility of the cellular amination network in *E. coli*" for consideration by *eLife*. Your article has been reviewed by 3 peer reviewers, one of whom is a member of our Board of Reviewing Editors, and the evaluation has been overseen by a Reviewing Editor and Gisela Storz as the Senior Editor. The following individuals involved in review of your submission have agreed to reveal their identity: Lawrence Reitzer (Reviewer #2); Daniel Segre (Reviewer #3).

Essential revisions:

1) Reviewers 1 and 3 provide many suggestions for improving the clarity of the text and figures.

2) Reviewer 2 highlights the need for expanded discussion of relevant topics, including metabolite levels, transaminases, regulation of enzymes, etc.

3) The authors are asked to address a lingering concern regarding the physiological relevance of the results, i.e. the fact that this result may be limited to a highly engineered strain and specific growth conditions and not observed in nature.

*Reviewer #1 (Recommendations for the authors):*

– Figure 1 is confusing to read as it is first referenced in the introduction, referring to amination on the right side of the figure, and then again referenced in the Results section, referring to amine donation routes on the left side. Figures are generally read from left to right, so simply inverting the figure will help bring the reader's attention to canonical ammonium assimilation first. Also, further distinguishing ammonium molecules and amine groups, possibly with different shapes, would be beneficial.

– Interchanging 2-ketoglutarate and 2-oxoglutarate in the text is confusing, as not all readers will recognize that they refer to the same molecule. Please pick one and stick with it.

– For making additional comparisons between strains and amine donors, it would be beneficial to see a table of calculated growth rates in the supplementary material, when relevant (proline supporting a higher growth rate than glutamate, line 165, for example). This may also support claims of certain amine donors or engineered strains being more efficient than others.

*Reviewer #3 (Recommendations for the authors):*

While the paper is very well written, is also fairly complex. Figure 1 is very useful to help the readers orient themselves. The numbers associated with the different paths through which amines could enter the network are also a helpful way of classifying the different modalities. What I would like to suggest is for the authors to refer more often and more explicitly to these modalities throughout the manuscript, by referring to those same four categories. I think this would help readers reconnect to Figure 1, especially when reaching the section on alternative routes of ammonium assimilation (page 14).

I am also wondering whether there is a way to display the overall relationship of the different amino acids to each other. I understand that it would be too complicated to display the complete metabolic network with the different amino acids. However, perhaps it is possible and useful to encode the relationship through a network of similarity of phenotypic behavior (e.g. draw a link between two amino acids if the amine assimilation behavior on those two amino acids is similar). This map could serve as an illustration of the overall structure of the dataset, and could reveal structural features difficult to infer from individual growth curves. I would invite the authors to consider generating such a map and potentially including it in the manuscript if it seems informative.

There are two broad questions that I found myself asking after reading this paper. They are both beyond the scope of the current paper, but I (and the readers) might be interested in hearing if you have thoughts you would be willing to share (e.g. in the discussion) on these points. The first question is whether you expect some or all of the patterns observed in *E. coli* to be broadly shared by other bacteria, or if – on the contrary – you think that radically different amination networks may be found in other bacteria (even closely related ones, which may have similar metabolism, but different regulatory networks) or other organisms. The second question is whether in your opinion the specific architecture you discovered can teach us something about the evolutionary history of nitrogen assimilation strategies.

---

## [Author Response]

Essential revisions:Reviewer #1 (Recommendations for the authors):– Figure 1 is confusing to read as it is first referenced in the introduction, referring to amination on the right side of the figure, and then again referenced in the Results section, referring to amine donation routes on the left side. Figures are generally read from left to right, so simply inverting the figure will help bring the reader's attention to canonical ammonium assimilation first. Also, further distinguishing ammonium molecules and amine groups, possibly with different shapes, would be beneficial.

We modified the figure accordingly to make it easier to read.

– Interchanging 2-ketoglutarate and 2-oxoglutarate in the text is confusing, as not all readers will recognize that they refer to the same molecule. Please pick one and stick with it.

We thank the reviewer for identifying this inconsistency. Now, all 2-oxoglutarates are changed 2-ketoglutarate.

– For making additional comparisons between strains and amine donors, it would be beneficial to see a table of calculated growth rates in the supplementary material, when relevant (proline supporting a higher growth rate than glutamate, line 165, for example). This may also support claims of certain amine donors or engineered strains being more efficient than others.

We agree with the reviewer that this information might be of interest to the reader. Correspondingly, we added a table to the supplementary material (Supplementary File S1).

Reviewer #3 (Recommendations for the authors):While the paper is very well written, is also fairly complex. Figure 1 is very useful to help the readers orient themselves. The numbers associated with the different paths through which amines could enter the network are also a helpful way of classifying the different modalities. What I would like to suggest is for the authors to refer more often and more explicitly to these modalities throughout the manuscript, by referring to those same four categories. I think this would help readers reconnect to Figure 1, especially when reaching the section on alternative routes of ammonium assimilation (page 14).

We thank the reviewer for the suggestion, which is in line with the suggestions of reviewer 1. We are now referring to the mechanisms from Figure 1 in the relevant sections of the manuscript.

I am also wondering whether there is a way to display the overall relationship of the different amino acids to each other. I understand that it would be too complicated to display the complete metabolic network with the different amino acids. However, perhaps it is possible and useful to encode the relationship through a network of similarity of phenotypic behavior (e.g. draw a link between two amino acids if the amine assimilation behavior on those two amino acids is similar). This map could serve as an illustration of the overall structure of the dataset, and could reveal structural features difficult to infer from individual growth curves. I would invite the authors to consider generating such a map and potentially including it in the manuscript if it seems informative.

According to the reviewer’s suggestion, we introduced a descriptive section in lines 486 – 493 and refer to a new supplementary Figure 2—figure supplement 1 (line 487).

There are two broad questions that I found myself asking after reading this paper. They are both beyond the scope of the current paper, but I (and the readers) might be interested in hearing if you have thoughts you would be willing to share (e.g. in the discussion) on these points. The first question is whether you expect some or all of the patterns observed in *E. coli* to be broadly shared by other bacteria, or if – on the contrary – you think that radically different amination networks may be found in other bacteria (even closely related ones, which may have similar metabolism, but different regulatory networks) or other organisms. The second question is whether in your opinion the specific architecture you discovered can teach us something about the evolutionary history of nitrogen assimilation strategies.

We thank the reviewer for this suggestion. We comment on the universal use of glutamate for ammonium fixation from line 536 and give some thoughts about the amination network (lines 547 – 554).